# Decoupling the Depth and Scope of Graph Neural Networks

**Hanqing Zeng**
USC
zengh@usc.edu

**Muhan Zhang**
Peking University, BIGAI
muhan@pku.edu.cn

**Yinglong Xia**
Facebook AI
yxia@fb.com

**Ajitesh Srivastava**
USC
ajiteshs@usc.edu

**Andrey Malevich**
Facebook AI
amalevich@fb.com

**Rajgopal Kannan**
US ARL
rajgopal.kannan.civ@mail.mil

**Viktor Prasanna**
USC
prasanna@usc.edu

**Long Jin**
Facebook AI
longjin@fb.com

**Ren Chen**
Facebook AI
renchen@fb.com

## Abstract

State-of-the-art Graph Neural Networks (GNNs) have limited scalability with respect to the graph and model sizes. On large graphs, increasing the model depth often means exponential expansion of the scope (*i.e.*, receptive field). Beyond just a few layers, two fundamental challenges emerge: 1. degraded *expressivity* due to oversmoothing, and 2. expensive *computation* due to neighborhood explosion. We propose a design principle to decouple the depth and scope of GNNs – to generate representation of a target entity (*i.e.*, a node or an edge), we first extract a localized subgraph as the *bounded-size* scope, and then apply a GNN of arbitrary depth on top of the subgraph. A properly extracted subgraph consists of a small number of critical neighbors, while excluding irrelevant ones. The GNN, no matter how deep it is, smooths the local neighborhood into informative representation rather than oversmoothing the global graph into "white noise". Theoretically, decoupling improves the GNN expressive power from the perspectives of graph signal processing (GCN), function approximation (GraphSAGE) and topological learning (GIN). Empirically, on seven graphs (with up to 110M nodes) and six backbone GNN architectures, our design achieves significant accuracy improvement with orders of magnitude reduction in computation and hardware cost.

## 1 Introduction

Graph Neural Networks (GNNs) have now become the state-of-the-art models for graph mining [48, 13, 58], facilitating applications such as social recommendation [35, 52, 37], knowledge understanding [40, 38, 59] and drug discovery [43, 32]. With the numerous architectures proposed [22, 12, 44, 49], it still remains an open question how to effectively scale up GNNs with respect to both the model size and graph size. There are two fundamental obstacles when we increase the number of GNN layers:

- *Expressivity* challenge (*i.e.*, oversmoothing [30, 36, 39, 17]): iterative mixing of neighbor features collapses embedding vectors of different nodes into a fixed, low-dimensional subspace.

---

Correspondence to: Muhan Zhang, muhan@pku.edu.cn

35th Conference on Neural Information Processing Systems (NeurIPS 2021).

- *Scalability* challenge (*i.e.*, neighbor explosion [7, 8, 9, 55]): recursive expansion of multi-hop neighborhood results in exponentially growing receptive field size (and thus computation cost).

To address the expressivity challenge, most remedies focus on neural architecture exploration: [44, 12, 49, 29] propose more expressive aggregation functions when propagating neighbor features. [50, 28, 18, 34, 1, 33, 31] use residue-style design components to construct flexible and dynamic receptive fields. Among them, [50, 28, 18] use skip-connection across multiple GNN layers, and [34, 1, 33, 31] encourage multi-hop message passing within each single layer. As for the scalability challenge, sampling methods have been explored to improve the training speed and efficiency. Importance based layer-wise sampling [8, 7, 61] and subgraph-based sampling [54, 9, 55] alleviate neighbor explosion, while preserving training accuracy. Unfortunately, such sampling methods cannot be naturally generalized to inference without accuracy loss (see also Section 4).

The above lines of research have only guided us to partial solutions. Yet what is the root cause of both the expressivity and scalability challenges? Setting aside the design of GNN architectures or sampling schemes, we provide an alternative perspective by interpretting the *data* in a different way.

**Two views on the graph.** Given an input graph $\mathcal{G}$ with node set $\mathcal{V}$, the most straightforward way to understand $\mathcal{G}$ is by viewing it as a single *global* graph. So any two nodes $u$ and $v$ belong to the same $\mathcal{G}$, and if $u$ and $v$ lie in the same connected component, they will ultimately see each other in their own neighborhood no matter how far away $u$ and $v$ are. Alternative to the above *global* view, we can take a *local* view on $\mathcal{G}$. For each node $v$, there is a *latent* $\mathcal{G}_{[v]}$ surrounding it which captures the characteristics of just $v$ itself. The full $\mathcal{G}$ is *observed* (by the data collection process) as the union of all such $\mathcal{G}_{[v]}$. Consequently, $\mathcal{V}_{[v]}$ rather than $\mathcal{V}$ defines $v$'s neighborhood: if $u \notin \mathcal{V}_{[v]}$, $v$ will never consider $u$ as a neighbor. Our "decoupling" design is based on the local view.

**Scope of GNNs.** Both the expressivity and scalability challenges are closely related to the enlargement of the GNN's scope (*i.e.*, receptive field). More importantly, how we define the scope is determined by how we view $\mathcal{G}$. With the global view above, an $L$-layer GNN has the scope of the full $L$-hop neighborhood. With the local view, the GNN scope is simply $\mathcal{V}_{[v]}$ regardless of the GNN depth. The two existing lines of research, one on architectural exploration and the other on sampling, both take the *global* view. Consequently, the depth (*i.e.*, number of layers) and scope of such GNNs are *tightly coupled*. Such coupling significantly limits the design space exploration of GNNs with various depths [53]. Consider the example of `ogbn-products`, a medium-scale graph in Open Graph Benchmark [16]. The average number of 4-hop neighbors is around 0.6M, corresponding to 25% of the full graph size. To generate representation of a single target node, a 4-layer coupled GNN needs to propagate features from the 0.6M neighbors. Such propagation can be inefficient or even harmful since most nodes in the huge neighborhood would be barely relevant to the target node.

**Decoupling the GNN depth and scope.** Taking the local view on $\mathcal{G}$, we propose a general design principle to decouple the GNN depth and scope. To generate the representation of the target node $v$, we first extract from $\mathcal{G}$ a small subgraph $\mathcal{G}_{[v]}$ surrounding $v$. On top of $\mathcal{G}_{[v]}$, we apply a GNN whose number of layers and message passing functions can be flexibly chosen. "Decoupling" means we treat the scope extraction function and GNN depth as two independently tuned parameters – effectively we introduce a new dimension in the GNN design space. We intuitively illustrate the benefits of decoupling by an example GNN construction, where the scope is the $L$-hop neighborhood and depth is $L'$ ($L' > L$). When we use more layers ($L'$) than hops ($L$), each pair of subgraph nodes may exchange messages multiple times. The extra message passing helps the GNN better absorb and embed the information within scope, and thus leads to higher expressivity. We further justify the above intuition with multifaceted theoretical analysis. From the graph signal processing perspective, we prove that decoupled-GCN performs local-smoothing rather than oversmoothing, as long as the scopes of different target nodes are different. From the function approximation perspective, we construct a linear target function on neighbor features and show that decoupling the GraphSAGE model reduces the function approximation error. From the topological learning perspective, we apply deep GIN-style message passing to differentiate non-regular subgraphs of a regular graph. As a result, our model is more powerful than the 1-dimensional Weisfeiler-Lehman test [41].

**Practical implementation: SHADOW-GNN.** The decoupling principle leads to a practical implementation, SHADOW-GNN: Decoupled GNN on a shallow subgraph. In SHADOW-GNN, the scope is a shallow yet informative subgraph, only containing a fraction of the 2- or 3-hop neighbors of $\mathcal{G}$ (see Section 5). On the other hand, the model of SHADOW-GNN is deeper (*e.g.*, $L' = 5$). To efficiently construct the shallow scope on commodity hardware, we propose various subgraph

extraction functions. To better utilize the subgraph node embeddings after deep message passing, we propose neural architecture extensions such as pooling and ensemble. Empirically, our "decoupling" design improves both the accuracy and scalability. On seven benchmarks (including the largest `ogbn-papers100M` graph with 111M nodes) and across two graph learning tasks, SHADOW-GNNs achieve significant accuracy gains compared to the original models. Meanwhile, the computation and hardware costs are reduced by orders of magnitude. Our code is available at `https://github.com/facebookresearch/shaDow_GNN`

## 2 Preliminaries

Let $\mathcal{G}(\mathcal{V}, \mathcal{E}, \boldsymbol{X})$ be an undirected graph, with node set $\mathcal{V}$, edge set $\mathcal{E} \subseteq \mathcal{V} \times \mathcal{V}$ and node feature matrix $\boldsymbol{X} \in \mathbb{R}^{|\mathcal{V}| \times d}$. Let $\mathcal{N}_v$ denote the set of $v$'s direct neighbors in $\mathcal{G}$. The $u^{\text{th}}$ row of $\boldsymbol{X}$ corresponds to the length-$d$ feature of node $u$. Let $\boldsymbol{A}$ be the adjacency matrix of $\mathcal{G}$ where $A_{u,v} = 1$ if edge $(u, v) \in \mathcal{E}$ and $A_{u,v} = 0$ otherwise. Let $\boldsymbol{D}$ be the diagonal degree matrix of $\boldsymbol{A}$. Denote $\widetilde{\boldsymbol{A}} = \boldsymbol{D}_*^{-\frac{1}{2}} \boldsymbol{A}_* \boldsymbol{D}_*^{-\frac{1}{2}}$ as the adjacency matrix after symmetric normalization ("$*$" means augmented with self-edges), and $\widehat{\boldsymbol{A}} = \boldsymbol{D}^{-1} \boldsymbol{A}$ (or $\boldsymbol{D}_*^{-1} \boldsymbol{A}_*$) as the one after random walk normalization. Let subscript "$[u]$" mark the quantities corresponding to a subgraph surrounding node $u$. For example, the subgraph itself is $\mathcal{G}_{[u]}$. Subgraph matrices $\boldsymbol{X}_{[v]}$ and $\boldsymbol{A}_{[v]}$ have the same dimension as the original $\boldsymbol{X}$ and $\boldsymbol{A}$. Yet, row vector $\left[\boldsymbol{X}_{[v]}\right]_u = \boldsymbol{0}$ for $u \notin \mathcal{V}_{[v]}$. Element $\left[\boldsymbol{A}_{[v]}\right]_{u,w} = 0$ if either $u \notin \mathcal{V}_{[v]}$ or $w \notin \mathcal{V}_{[v]}$. For an $L$-layer GNN, let superscript "$(\ell)$" denote the layer-$\ell$ quantities. Let $d^{(\ell)}$ be the number of channels for layer $\ell$; $\boldsymbol{H}^{(\ell-1)} \in \mathbb{R}^{|\mathcal{V}| \times d^{(\ell-1)}}$ and $\boldsymbol{H}^{(\ell)} \in \mathbb{R}^{|\mathcal{V}| \times d^{(\ell)}}$ be the input and output features. So $\boldsymbol{H}^{(0)} = \boldsymbol{X}$ and $d^{(0)} = d$. Further, let $\sigma$ be the activation and $\boldsymbol{W}^{(\ell)}$ be the learnable weight. For example, a GCN layer performs $\boldsymbol{H}^{(\ell)} = \sigma\left(\widetilde{\boldsymbol{A}} \boldsymbol{H}^{(\ell-1)} \boldsymbol{W}^{(\ell)}\right)$. A GraphSAGE layer performs $\boldsymbol{H}^{(\ell)} = \sigma\left(\boldsymbol{H}^{(\ell-1)} \boldsymbol{W}_1^{(\ell)} + \widehat{\boldsymbol{A}} \boldsymbol{H}^{(\ell-1)} \boldsymbol{W}_2^{(\ell)}\right)$.

Our analysis in Section 3 mostly focuses on the node classification task. Yet our design can be generalized to the link prediction task, as demonstrated by our experiments in Section 5.

**Definition 2.1.** *(Depth of subgraph) Assume the subgraph $\mathcal{G}_{[v]}$ is connected. The depth of $\mathcal{G}_{[v]}$ is defined as $\max_{u \in \mathcal{V}_{[v]}} d(u, v)$, where $d(u, v)$ denotes the shortest path distance from $u$ to $v$.*

The above definition enables us to make comparison such as "the GNN is deeper than the subgraph". For "decoupling the depth and scope", we refer to the model depth rather than the subgraph depth.

## 3 Decoupling the Depth and Scope of GNNs

"Decoupling the depth and scope of GNNs" is a design principle to improve the expressivity and scalability of GNNs without modifying the layer architecture. We name a GNN after decoupling a SHADOW-GNN (see Section 3.6 for explanation of the name). Compared with a normal GNN, SHADOW-GNN contains an additional component: the subgraph extractor `EXTRACT`. To generate embedding of a target node $v$, SHADOW-GNN proceeds as follows: 1. We use `EXTRACT`$(v, \mathcal{G})$ to return a connected $\mathcal{G}_{[v]}$, where $\mathcal{G}_{[v]}$ is a subgraph containing $v$, and the depth of $\mathcal{G}_{[v]}$ is $L$. 2. We build an $L'$-layer GNN on $\mathcal{G}_{[v]}$ by treating $\mathcal{G}_{[v]}$ as the *new* full graph and by ignoring all nodes / edges not in $\mathcal{G}_{[v]}$. So $\mathcal{G}_{[v]}$ is the scope of SHADOW-GNN. The key point reflecting "decoupling" is that $L' > L$.

A normal GNN is closely related to a SHADOW-GNN. Under the normal setup, an $L$-layer GNN operates on the full $\mathcal{G}$ and propagates the influence from all the neighbors up to $L$ hops away from $v$. Such a GNN is equivalent to a model where `EXTRACT` returns the full $L$-hop subgraph and $L' = L$.

We theoretical demonstrate how SHADOW-GNN improves expressivity from three different angles. On SHADOW-GCN (Section 3.1), we come from the *graph signal processing* perspective. The GCN propagation can be interpreted as applying filtering on the node signals [47]. Deep models correspond to high-pass filters. Filtering the local graph $\mathcal{G}_{[v]}$ preserves richer information than the global $\mathcal{G}$. On SHADOW-SAGE (Section 3.2), we view the GNN as a *function approximator*. We construct a target function and study how decoupling reduces the approximation error. On SHADOW-GIN (Section 3.3), we focus on learning *topological information*. We show that decoupling helps capture local graph structure which the 1D Weisfeiler-Lehman test fails to capture.

## 3.1 Expressivity Analysis on SHADOW-GCN: Graph Signal Processing Perspective

GCNs [22] suffer from "oversmoothing" [30] – Each GCN layer smooths the features of the direct (*i.e.*, 1-hop) neighbors, and many GCN layers smooths the features of the full graph. Eventually, such repeated smoothing process propagates to any target node just the averaged feature of all $\mathcal{V}$. "Oversmoothing" thus incurs significant information loss by wiping out all local information.

Formally, suppose the original features $\boldsymbol{X}$ reside in a high-dimensional space $\mathbb{R}^{|\mathcal{V}| \times d}$. Oversmoothing pushes $\boldsymbol{X}$ towards a low-dimensional subspace $\mathbb{R}^{|\mathcal{V}| \times d'}$, where $d' < d$. Corresponding analysis comes from two perspectives: oversmoothing by *a deep GCN*, and oversmoothing by *repeated GCN-style propagation*. The former considers the full neural network with non-linear activation, weight and bias. The later characterizes the aggregation matrix $\boldsymbol{M} = \lim_{L \to \infty} \widetilde{\boldsymbol{A}}^L \boldsymbol{X}$. It is shown that even with the vanilla architecture, a deep GCN with bias parameters does *not* oversmooth [17]. In addition, various tricks [60, 39, 34] can prevent oversmoothing from the neural network perspective. However, a deep GCN still suffers from accuracy drop, indicating that the GCN-style propagation (rather than other GCN components like activation and bias) may be the fundamental reason causing difficulty in learning. Therefore, we study the asymptotic behavior of the aggregation matrix $\boldsymbol{M}$ under the normal and SHADOW design. In other words, here in Section 3.1, we ignore the non-linear activation and bias parameters. Such setup is consistent with many existing literature such as [30, 33, 34, 60].

**Proposition 3.1.** $\infty$ *number of feature propagation by* SHADOW-GCN *leads to*

$$\boldsymbol{m}_{[v]} = \left[ e_{[v]} \right]_v \cdot \left( \boldsymbol{e}_{[v]}^{\mathsf{T}} \boldsymbol{X}_{[v]} \right) \tag{1}$$

*where* $\boldsymbol{e}_{[v]}$ *is defined by* $\left[ e_{[v]} \right]_u = \sqrt{\frac{\delta_{[v]}(u)}{\sum_{w \in \mathcal{V}_{[v]}} \delta_{[v]}(w)}}$; $\delta_{[v]}(u)$ *returns the degree of* $u$ *in* $\mathcal{G}_{[v]}$ *plus 1.*

**Oversmoothing by normal GCN propagation.** With a large enough $L$, the full $L$-hop neighborhood becomes $\mathcal{V}$ (assuming connected $\mathcal{G}$). So $\forall\, u, v$, we have $\mathcal{G}_{[u]} = \mathcal{G}_{[v]} = \mathcal{G}$, implying $\boldsymbol{e}_{[u]} = \boldsymbol{e}_{[v]}$ and $\boldsymbol{X}_{[u]} = \boldsymbol{X}_{[v]} = \boldsymbol{X}$. From Proposition 3.1, the aggregation converges to a point where *no* feature and *little* structural information of the target is preserved. The only information in $\boldsymbol{m}_{[v]}$ is $v$'s degree.

**Local-smoothing by SHADOW-GCN propagation.** With a fixed subgraph, no matter how many times we aggregate using $\widetilde{\boldsymbol{A}}_{[v]}$, the layers will not include the faraway irrelevant nodes. From Proposition 3.1, $\boldsymbol{m}_{[v]}$ is a linear combination of the neighbor features $\boldsymbol{X}_{[v]}$. Increasing the number of layers only pushes the *coefficients* of each neighbor features to the stationary values. The *domain* $\boldsymbol{X}_{[v]}$ of such linear transformation is solely determined by EXTRACT and is independent of the model depth. Intuitively, if EXTRACT picks non-identical subgraphs for two nodes $u$ and $v$, the aggregations should be different due to the different domains of the linear transformation. Therefore, SHADOW-GCN preserves *local* feature information whereas normal GCN preserves *none*. For structural information in $\boldsymbol{m}_{[v]}$, note that $\boldsymbol{e}_{[v]}$ is a normalized degree distribution of the subgraph around $v$, and $\left[ e_{[v]} \right]_v$ indicates the role of the target node in the subgraph. By simply letting EXTRACT return the 1-hop subgraph, $\left[ e_{[v]} \right]_v$ alone already contains all the information preserved by a normal GCN, which is $v$'s degree in $\mathcal{G}$. For the general EXTRACT, $\boldsymbol{e}_{[v]}$ additionally reflects $v$'s ego-net structure. Thus, a deep SHADOW-GCN preserves more structural information than a deep GCN.

**Theorem 3.2.** *Let* $\overline{\boldsymbol{m}}_{[v]} = \phi_{\mathcal{G}}(v) \cdot \boldsymbol{m}_{[v]}$ *where* $\phi_{\mathcal{G}}$ *is any non-zero function only depending on the structural property of* $v$. *Let* $\mathcal{M} = \{ \overline{\boldsymbol{m}}_{[v]} \mid v \in \mathcal{V} \}$. *Given* $\mathcal{G}$, EXTRACT *and some continuous probability distribution in* $\mathbb{R}^{|\mathcal{V}| \times d}$ *to generate* $\boldsymbol{X}$, *then* $\overline{\boldsymbol{m}}_{[v]} \neq \overline{\boldsymbol{m}}_{[u]}$ *if* $\mathcal{V}_{[u]} \neq \mathcal{V}_{[v]}$, *almost surely.*

**Corollary 3.2.1.** *Consider* EXTRACT$_1$, *where* $\forall v \in \mathcal{V}$, $\left| \mathcal{V}_{[v]} \right| \leq n$. *Then* $|\mathcal{M}| \geq \left\lceil \frac{|\mathcal{V}|}{n} \right\rceil$ *a.s.*

**Corollary 3.2.2.** *Consider* EXTRACT$_2$, *where* $\forall\, u, v \in \mathcal{V}$, $\mathcal{V}_{[v]} \neq \mathcal{V}_{[u]}$. *Then* $|\mathcal{M}| = |\mathcal{V}|$ *a.s.*

Theorem 3.2 proves SHADOW-GCN does not oversmooth: 1. A normal GCN pushes the aggregation of same-degree nodes to the same point, while SHADOW-GCN with EXTRACT$_2$ ensures any two nodes (even with the same degree) have different aggregation. 2. A normal GCN wipes out all information in $\boldsymbol{X}$ after many times of aggregation, while SHADOW-GCN always preserves feature information. Particularly, with $\phi_{\mathcal{G}}(v) = \left( \delta_{[v]}(v) \right)^{-1/2}$, a normal GCN generates only one unique value of $\overline{\boldsymbol{m}}$ for all $v$. By contrast, SHADOW-GNN generates $|\mathcal{V}|$ different values for any $\phi_{\mathcal{G}}$ function.

## 3.2 Expressivity Analysis on SHADOW-SAGE: Function Approximation Perspective

We compare the expressivity by showing 1. SHADOW-SAGE can express all functions GraphSAGE can, and 2. SHADOW-SAGE can express some function GraphSAGE cannot. Recall, a GraphSAGE layer performs the following: $h_v^{(\ell)} = \sigma\left(\left(W_1^{(\ell)}\right)^\mathsf{T} h_v^{(\ell-1)} + \left(W_2^{(\ell)}\right)^\mathsf{T}\left(\frac{1}{|\mathcal{N}_v|}\sum_{u\in\mathcal{N}_v} h_u^{(\ell-1)}\right)\right)$. We can prove Point 1 by making an $L'$-layer SHADOW-SAGE identical to an $L$-layer GraphSAGE with the following steps: 1. let EXTRACT return the full $L$-hop neighborhood, and 2. set $W_1^{(\ell)} = I$, $W_2^{(\ell)} = 0$ for $L+1 \leq \ell \leq L'$. For point 2, we consider a target function: $\tau\left(X, \mathcal{G}_{[v]}\right) = C \cdot \sum_{u\in\mathcal{V}_{[v]}} \delta_{[v]}(u) \cdot x_u$ for some neighborhood $\mathcal{G}_{[v]}$, scaling constant $C$ and $\delta_{[v]}(u)$ as defined in Proposition 3.1. An expressive model should be able to learn well this simple linear function $\tau$.

GraphSAGE cannot learn $\tau$ accurately, while SHADOW-SAGE can. We first show the GraphSAGE case. Let the depth of $\mathcal{G}_{[v]}$ be $L$. Firstly, we need GraphSAGE to perform message passing for exactly $L$ times (where such a model can be implemented by, *e.g.*, $L$ layers or $L'$ layers with $W_2 = 0$ for $L' - L$ layers). Otherwise, the extra $L' - L$ message passings will propagate influence from nodes $v' \notin \mathcal{V}_{[v]}$, violating the condition that $\tau$ is independent of $v'$. Next, suppose GraphSAGE can learn a function $\zeta$ such that on some $\mathcal{G}'_{[v]}$, we have $\zeta\left(\mathcal{G}'_{[v]}\right) = \tau\left(\mathcal{G}'_{[v]}\right)$. We construct another $\mathcal{G}''_{[v]}$ by adding an extra edge $e$ connecting two depth-$L$ nodes in $\mathcal{G}'_{[v]}$. Edge $e$ changes the degree distribution $\delta_{[v]}(\cdot)$, and thus $\tau\left(\mathcal{G}'_{[v]}\right) \neq \tau\left(\mathcal{G}''_{[v]}\right)$. On the other hand, there is no way for GraphSAGE to propagate the influence of edge $e$ to the target $v$, unless the model performs at least $L+1$ message passings. So $\zeta\left(\mathcal{G}'_{[v]}\right) = \zeta\left(\mathcal{G}''_{[v]}\right)$ regardless of the activation function and weight parameters. Therefore, $\zeta \neq \tau$.

For SHADOW-SAGE, let EXTRACT return $\mathcal{G}_{[v]}$. Then the model can output $\zeta' = \left[\widehat{A}_{[v]}^{L'} X\right]_{v,:}$ after we 1. set $W_1^{(\ell)} = 0$ and $W_2^{(\ell)} = I$ for all layers, and 2. either remove the non-linear activation or bypass ReLU by shifting $X$ with bias. With known results in Markov chain convergence theorem [27], we derive the following theorem by analyzing the convergence of $\widehat{A}_{[v]}^{L'}$ when $L' \to \infty$.

**Theorem 3.3.** SHADOW-SAGE *can approximate $\tau$ with error decaying exponentially with depth.*

We have the following conclusions from above: 1. SHADOW-SAGE is more expressive than Graph-SAGE. 2. appropriate EXTRACT function improves SHADOW-GNN expressivity, 3. There exists cases where it may be desirable to set the SHADOW-GNN depth much larger than the subgraph depth.

## 3.3 Expressivity Analysis on SHADOW-GIN: Topological Learning Perspective

While GCN and GraphSAGE are popular architectures in practice, they are not the theoretically most discriminative ones. The work in [49] establishes the relation in discriminativeness between GNNs and 1-dimensional Weisfeiler-Lehman test (*i.e.*, 1-WL). And GIN [49] is an example architecture achieving the same discriminativeness as 1-WL. We show that applying the decoupling principle can further improve the discriminativeness of such GNNs, making them more powerful than 1-WL.

1-WL is a graph isomorphism test aiming at distinguishing graphs of different structures. A GNN as expressive as 1-WL thus well captures the topological property of the target node. While 1-WL is already very powerful, it may still fail in some cases. *e.g.*, it cannot distinguish certain non-isomorphic, *regular* graphs. To understand why SHADOW-GNN works, we first need to understand why 1-WL fails. In a regular graph, all nodes have the same degree, and thus the "regular" property describes a *global* topological symmetry among nodes. Unfortunately, 1-WL (and the corresponding normal GNN) also operates *globally* on $\mathcal{G}$. Intuitively, on two different regular graphs, there is no way for 1-WL (and the normal GNN) to assign different labels by breaking such symmetry.

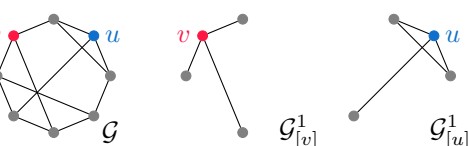

Figure 1: Example 3-regular graph and the 1-hop subgraphs of the target nodes $u$ and $v$.

On the other hand, SHADOW-GNN can break such symmetry by applying decoupling. In Section 1, we have discussed how SHADOW-GNN is built from the *local* perspective on the full graph. The

key property benefiting SHADOW-GNN is that *a subgraph of a regular graph may not be regular*. Thus, SHADOW-GNN can distinguish nodes in a regular graph with the non-regular subgraphs as the scope. We illustrate the intuition with the example in Figure 1. The graph $\mathcal{G}$ is 3-regular and we assume all nodes have identical features. Our goal is to discriminate nodes $u$ and $v$ since their neighborhood structures are different. No matter how many iterations 1-WL runs, or how many layers the normal GNN has, they cannot distinguish $u$ and $v$. On the other hand, a SHADOW-GNN with 1-hop EXTRACT and at least 2 layers can discriminate $u$ and $v$.

**Theorem 3.4.** *Consider GNNs whose layer function is defined by*

$$\boldsymbol{h}_v^{(\ell)} = f_1^{(\ell)}\left(\boldsymbol{h}_v^{(\ell-1)}, \sum_{u\in\mathcal{N}_v} f_2^{(\ell)}\left(\boldsymbol{h}_v^{(\ell-1)}, \boldsymbol{h}_u^{(\ell-1)}\right)\right), \tag{2}$$

*where $f_1^{(\ell)}$ and $f_2^{(\ell)}$ are the update and message functions of layer-$\ell$, implemented as MLPs. Then, such SHADOW-GNN is more discriminative than the 1-dimensional Weisfeiler-Lehman test.*

The theorem also implies that SHADOW-GIN is more discriminative than a normal GIN due to the correspondence between GIN and 1-WL. See Appendix A for the proof of all theorems in Section 3.

## 3.4 Subgraph Extraction Algorithms

Our decoupling principle does not rely on specific subgraph extraction algorithms. Appropriate EXTRACT can be customized given the characteristic of $\mathcal{G}$, and different EXTRACT leads to different *implementation* of our decoupling principle. In general, we summarize three approaches to design EXTRACT: 1. *heuristic based*, where we pick graph metrics that reflect neighbor importance and then design EXTRACT by such metrics; 2. *model based*, where we assume a generation process on $\mathcal{G}$ and set EXTRACT as the reverse process, and 3. *learning based*, where we integrate the design of EXTRACT as part of the GNN training. In the following, we present several examples on heuristic based EXTRACT, which we also empirically evaluate in Section 5. We leave detailed evaluation on the model based and learning based EXTRACT as future work. See also Appendix C for details.

**Example heuristic based EXTRACT.** The algorithm is derived from the selected graph metrics. For example, with the metric being shortest path distance, we design a $L$-hop extractor. *i.e.*, EXTRACT returns the full set or randomly selected subset of the target node's $L$-hop neighbors in $\mathcal{G}$. Picking the random walk landing probability as the metric, we can design a PPR-based extractor. *i.e.*, we first run the Personalized PageRank (PPR) algorithm on $\mathcal{G}$ to derive the PPR score of other nodes relative to the target node. Then EXTRACT define $\mathcal{V}_{[v]}$ by picking the top-$K$ nodes with the highest PPR scores. The subgraph $\mathcal{G}_{[v]}$ is the *node-induced subgraph*[1] of $\mathcal{G}$ from $\mathcal{V}_{[v]}$. One can easily extend this approach by using other metrics such as Katz index [20], SimRank [19] and feature similarity.

## 3.5 Architecture

**Subgraph pooling.** For a normal GNN performing node classification, the multi-layer message passing follows a "tree structure". The nodes at level $L$ of the tree correspond to the $L$-hop neighborhood. And the tree root outputs the final embedding of the target node. Thus, there is no way to apply subgraph pooling or READOUT on the final layer output, since the "pool" only contains a single vector. For a SHADOW-GNN, since we decouple the $L^{\text{th}}$ layer from the $L$-hop neighborhood, it is natural to let each layer (including the final layer) output embeddings for all subgraph nodes. This leads to the design to READOUT all the subgraph node embeddings as the target node embedding.

We can understand the pooling for SHADOW-GNN from another perspective. In a normal GNN, the target node at the final layer receives messages from all neighbors, but two neighbor nodes may not have a chance to exchange any message to each other (*e.g.*, two nodes $L$-hop away from the target may be $2L$-hop away from each other). In our design, a SHADOW-GNN can pass messages between *any* pair of neighbors when the model depth is large enough. Therefore, all the subgraph node embeddings at the final layer capture meaningful information of the neighborhood.

In summary, the power of the decoupling principle lies in that *it establishes the connection between the node- / link-level task and the graph-level task. e.g.*, to classify a node is seen as to classify

---

[1]Unlike other PPR-based models [23, 6] which rewire the graph by treating top PPR nodes as direct neighbors, our PPR neighborhood preserves the original multi-hop topology by returning node-induced subgraph.

the subgraph surrounding the node. From the neural architecture perspective, we can apply any subgraph pooling / READOUT operation originally designed for graph classification (*e.g.*, [57, 24, 4]) to enhance the node classification / link prediction of SHADOW-GNN. In particular, in the vanilla SHADOW-GNN, we can implement a trivial READOUT as "discarding all neighbor embeddings", corresponding to performing center pooling. See Appendix D and F.3 for algorithm and experiments.

**Subgraph ensemble.** It may be challenging in practice to design a single EXTRACT capturing all meaningful characteristics of the neighborhood. We can use multiple EXTRACT to jointly define the receptive field, and then ensemble multiple SHADOW-GNN at the subgraph level. Consider $R$ candidates $\{\texttt{EXTRACT}_i\}$, each returning $\mathcal{G}^i_{[v]}$. To generate $v$'s embedding, we first use $R$ branches of $L'$-layer GNN to obtain intermediate embeddings for each $\mathcal{G}^i_v$, and then aggregate the $R$ embeddings by some learnable function $g$. In practice, we design $g$ as an attention based aggregation function (see Appendix D.2). Subgraph ensemble is useful both when $\{\texttt{EXTRACT}_i\}$ consists of different algorithms and when each $\texttt{EXTRACT}_i$ performs the same algorithm under different parameters.

CASE STUDY   Consider PPR-based $\texttt{EXTRACT}_i$ with different threshold $\theta_i$ on the neighbor PPR score. A SHADOW-GNN-ensemble can approximate PPRGo [6]. PPRGo generates embedding as: $\boldsymbol{\xi}_v = \sum_{u \in \mathcal{V}_{[v]}} \pi_u \boldsymbol{h}_v$, where $\pi_u$ is $u$'s PPR score and $\boldsymbol{h}_v = \texttt{MLP}\left(\boldsymbol{x}_v\right)$. We can partition $\mathcal{V}_{[v]} = \bigcup_{i=1}^{R} \mathcal{V}^i_{[v]}$ s.t. nodes in $\mathcal{V}^i_{[v]}$ have similar PPR scores denoted by $\widetilde{\pi}_i$, and $\widetilde{\pi}_i \leq \widetilde{\pi}_{i+1}$. So $\boldsymbol{\xi}_v = \sum_{i=1}^{R} \rho_i \left(\sum_{u \in \mathcal{V}'_i} \boldsymbol{h}_u\right)$, where $\rho_i = \widetilde{\pi}_i - \sum_{j < i} \widetilde{\pi}_j$ and $\mathcal{V}'_i = \bigcup_{k=i}^{R} \mathcal{V}^k_{[v]}$. Now for each branch of SHADOW-GNN-ensemble, let parameter $\theta_i = \widetilde{\pi}_i$ so that $\texttt{EXTRACT}_i$ returns $\mathcal{V}'_i$. The GNN on $\mathcal{V}'_i$ can then learn $\sum_{u \in \mathcal{V}'_i} \boldsymbol{h}_u$ (*e.g.*, by a simple "mean" READOUT). Finally, set the ensemble weight as $\rho_i$. SHADOW-GNN-ensemble learns $\boldsymbol{\xi}_v$. As EXTRACT also preserves graph topology, our model can be more expressive than PPRGo.

### 3.6   Practical Design: SHADOW-GNN

We now discuss the practical implementation of decoupled GNN – SHADOW-GNN. As the name suggests, in SHADOW-GNN, the scope is a shallow subgraph (*i.e.*, with depth often set to 2 or 3).

In many realistic scenarios (*e.g.*, citation networks, social networks, product recommendation graphs), a shallow neighborhood is both *necessary* and *sufficient* for the GNN to learn well. On "sufficiency", we consider the social network example: the friend of a friend of a friend may share little commonality with you, and close friends may be at most 2 hops away. Formally, by the $\gamma$-decaying theorem [56], a shallow neighborhood is sufficient to accurately estimate various graph metrics. On "necessity", since the neighborhood size may grow exponentially with hops, a deep neighborhood would be dominated by nodes irrelevant to the target. The corresponding GNN would first need to differentiate the many useless nodes from the very few useful ones, before it can extract meaningful features from the useful nodes. Finally, a shallow subgraph ensures scalability by avoiding "neighborhood explosion".

**Remark on decoupling.** So far we have defined a decoupled model as having the model depth $L'$ larger than the subgraph depth $L$. Strictly speaking, a decoupled model also admits $L' = L$. For example, suppose in the full $L$-hop neighborhood, there are 70% nodes $L$ hops away. Applying decoupling, the EXTRACT excludes most of the $L$-hop neighbors, and the resulting subgraph $\mathcal{G}_{[v]}$ contains only 20% nodes $L$ hops away. Then it is reasonable to consider an $L$-layer model on such a depth-$L$ subgraph as also a decouple model. Compared with an $L$-layer model on the full $L$-hop neighborhood, an $L$-layer model on such a depth-$L$ $\mathcal{G}_{[v]}$ propagates much less information from nodes $L$ hops away. So the $L$ message passings are indeed decoupled from the full $L$-hop neighborhood.

**Remark on neighborhood.** The "sufficiency" and "necessity" in shallow neighborhood are not universal. In many other applications, long-range dependencies can be critical, as studied in [2]. In such cases, our practical implementation of SHADOW-GNN would incur accuracy loss. However, our decoupling principle in general may still be beneficial – "shallow subgraph" is a practical guideline rather than a theoretical requirement. We leave the study on such applications as future work.

## 4   Related Work

**Deep GNNs.** To improve the GNN performance while increasing the model depth, various layer architectures have been proposed. AS-GCN [18], DeepGCN [28], JK-net [50], MixHop [1], Snowball [33], DAGNN [31] and GCNII [34] all include some variants of residue connection, either

across multiple layers or within a single layer. In principle, such architectures can also benefit the feature propagation of a deep SHADOW-GNN, since their design does not rely on a specific neighborhood (*e.g.*, $L$-hop). In addition to architectures, DropEdge [39] and Bayesian-GDC [14] propose regularization techniques by adapting dropout [42] to graphs. Such techniques are only applied during training, and inference may still suffer from issues such as oversmoothing.

**Sampling based methods.** Neighbor or subgraph sampling techniques have been proposed to improve training efficiency. FastGCN [8], VR-GCN [7], AS-GCN [18], LADIES [61] and MVS-GNN [10] sample neighbor nodes per GNN layer. Cluster-GCN [9] and GraphSAINT [55] sample a subgraph as the training minibatch. While sampling also changes the receptive field, all the above methods are fundamentally different from ours. The training samplers aim at estimating the quantities related to the full graph (*e.g.*, the aggregation of the full $L$-hop neighborhood), and so the inference model still operates on the full neighborhood to avoid accuracy loss. For SHADOW-GNN, since the decoupling principle is derived from a local view on $\mathcal{G}$, our EXTRACT does not estimate any full neighborhood quantities. Consequently, the sampling based methods only improve the training efficiency, while SHADOW-GNN addresses the computation challenge for both training and inference.

**Re-defining the neighborhood.** Various works reconstruct the original graph and apply the GNN on the re-defined neighborhood. GDC [23] views the reconstructed adjacency matrix as the diffusion matrix. SIGN [11] applies reconstruction for customized graph operators. AM-GCN [46] utilizes the reconstruction to separate feature and structure information. The above work re-define the neighborhood. However, they still have tightly coupled depth and scope. SHADOW-GNN can also work with the reconstructed graph $\mathcal{G}'$ by simply applying EXTRACT on $\mathcal{G}'$.

## 5 Experiments

**Setup.** We evaluate SHADOW-GNN on seven graphs. Six of them are for the node classification task: `Flickr` [55], `Reddit` [12], `Yelp` [55], `ogbn-arxiv`, `ogbn-products` and `ogbn-papers100M` [16]. The remaining is for the link prediction task: `ogbl-collab` [16]. The sizes of the seven graphs range from 9K nodes (`Flickr`) to 110M nodes (`ogbn-papers100M`). `Flickr`, `Reddit` and `Yelp` are under the inductive setting. `ogbn-arxiv`, `ogbn-products` and `ogbn-papers100M` are transductive. Consistent with the original papers, for the graphs on node classification, we measure "F1-micro" score for `Yelp` and "accuracy" for the remaining five graphs. For the link prediction task, we use "Hits@50" as the metric. See Appendix E.1 for details. We use W&B [5] for experiment tracking.

We construct SHADOW with six backbone models: GCN [22], GraphSAGE [12], GAT [44], JK-Net [50], GIN [49], SGC [47]. The first five are representatives of the state-of-the-art GNN architectures, jointly covering various message aggregation functions as well as the skip connection design. SGC simplifies normal GCN by moving all the neighbor propagation to the pre-processing step. Therefore, SGC is suitable for evaluating oversmoothing. The non-SHADOW models are trained with both full-batch and GraphSAINT-minibatch [55]. Due to the massive size of the full $L$-hop neighborhood, we need to perform sampling when training normal GNNs in order to make the computation time tractable. GraphSAINT is suitable for our purpose since 1. it is the state-of-the-art minibatch method which achieves high accuracy, and 2. it supports various GNN architectures and scales well to large graphs. On the other hand, for SHADOW-GNN, both training and inference are always executed in minibatches. One advantage of SHADOW-GNN is that the decoupling enables straightforward minibatch construction: each target just independently extracts the small subgraph on its own.

We implement two EXTRACT described in Section 3.4: 1. "PPR", where we set the node budget $K$ as $\{200, 400\}$ for the largest `ogbn-papers100M` and $K \leq 200$ for all other graphs; 2. "$L$-hop", where we set the depth as $\{1, 2\}$. We implement various subgraph pooling functions: "mean" and "max" evaluated in this section and others evaluated in Appendix F.3. For the model depth, since $L' = 3$ is the standard setting in the literature (*e.g.*, see the benchmarking in OGB [16]), we start from $L' = 3$ and further evaluate a deeper model of $L' = 5$. All accuracy are measured by 5 runs without fixing random seeds. Hyperparameter tuning and architecture configurations are in Appendix E.4.

**SHADOW-GNN neighborhood.** For both normal and SHADOW GNNs, Figure 2 shows on average how many neighbors are $L$ hops away from the target. For a normal GNN, the size of the neighborhood grows rapidly with respect to $L$, and the nodes 4 hops away dominate the neighborhood. For SHADOW-GNN using the Table 1 EXTRACT, most neighbors concentrate within 2 hops. A small number of nodes are 3 hops away. Almost no nodes are 4 or more hops away. Importantly, not

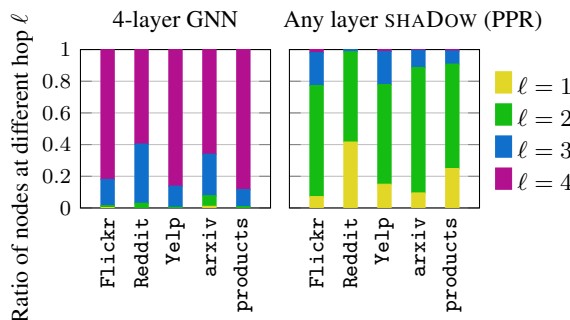

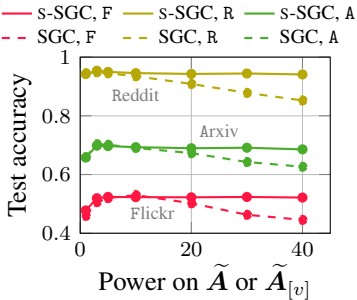

Figure 2: Neighborhood composition

Figure 3: SGC oversmoothing

only does the composition of the two kinds of neighborhood differ significantly, but also the size of SHADOW-GNN scope is much smaller (see also Table 1). Such small subgraphs are essential to high computation efficiency. Finally, we can ignore the very few distant neighbors ($L \geq 4$), and regard the (effective) depth of SHADOW-GNN subgraph as $L = 2$ (or at most 3). Under such practical value of $L$, a model with $L' \geq 3$ is indeed a SHADOW-GNN (see "Remark on decoupling" in Section 3.6).

Table 1: Comparison on test accuracy / F1-micro score and inference cost (tuned with DropEdge)

| Method | Layers | Flickr | | Reddit | | Yelp | | ogbn-arxiv | | ogbn-products | |
|---|---|---|---|---|---|---|---|---|---|---|---|
| | | Accuracy | Cost | Accuracy | Cost | F1-micro | Cost | Accuracy | Cost | Accuracy | Cost |
| GCN | 3 | 0.5159±0.0017 | 2E0 | 0.9532±0.0003 | 6E1 | 0.4028±0.0019 | 2E1 | 0.7170±0.0026 | 1E1 | 0.7567±0.0018 | 5E0 |
| | 5 | 0.5217±0.0016 | 2E2 | 0.9495±0.0012 | 1E3 | OOM | 1E3 | 0.7186±0.0017 | 1E3 | OOM | 9E2 |
| GCN-SAINT | 3 | 0.5155±0.0027 | 2E0 | 0.9523±0.0003 | 6E1 | 0.5110±0.0012 | 2E1 | 0.7093±0.0003 | 1E1 | **0.8003**±0.0024 | 5E0 |
| | 5 | 0.5165±0.0026 | 2E2 | 0.9523±0.0012 | 1E3 | 0.5012±0.0021 | 1E3 | 0.7039±0.0020 | 1E3 | 0.7992±0.0021 | 9E2 |
| SHADOW-GCN | 3 | 0.5234±0.0009 | **(1)** | 0.9576±0.0005 | **(1)** | 0.5291±0.0020 | **(1)** | 0.7180±0.0024 | **(1)** | 0.7742±0.0037 | **(1)** |
| (PPR) | 5 | 0.5268±0.0008 | 1E0 | 0.9564±0.0004 | 1E0 | 0.5323±0.0020 | 2E0 | 0.7206±0.0025 | 2E0 | 0.7821±0.0043 | 2E0 |
| +Pooling | 3/5 | **0.5286**±0.0013 | 1E0 | **0.9624**±0.0002 | 1E0 | **0.5393**±0.0036 | 2E0 | 0.7223±0.0018 | 2E0 | 0.7914±0.0044 | 2E0 |
| GraphSAGE | 3 | 0.5140±0.0014 | 3E0 | 0.9653±0.0002 | 5E1 | 0.6178±0.0033 | 2E1 | 0.7192±0.0027 | 1E1 | 0.7858±0.0013 | 4E0 |
| | 5 | 0.5154±0.0052 | 2E2 | 0.9626±0.0004 | 1E3 | OOM | 2E3 | 0.7193±0.0037 | 1E3 | OOM | 1E3 |
| SAGE-SAINT | 3 | 0.5176±0.0032 | 3E0 | 0.9671±0.0004 | 5E1 | 0.6453±0.0011 | 2E1 | 0.7107±0.0003 | 1E1 | 0.7923±0.0023 | 4E0 |
| | 5 | 0.5201±0.0032 | 2E2 | 0.9670±0.0010 | 1E3 | 0.6394±0.0003 | 2E3 | 0.7013±0.0021 | 1E3 | 0.7964±0.0022 | 1E3 |
| SHADOW-SAGE | 3 | 0.5312±0.0019 | 1E0 | 0.9672±0.0003 | 1E0 | 0.6542±0.0002 | 1E0 | 0.7163±0.0028 | 1E0 | 0.7935±0.0031 | 1E0 |
| (2-hop) | 5 | 0.5335±0.0015 | 2E0 | 0.9675±0.0005 | 2E0 | 0.6525±0.0003 | 2E0 | 0.7180±0.0012 | 2E0 | 0.7995±0.0022 | 2E0 |
| SHADOW-SAGE | 3 | 0.5356±0.0013 | **(1)** | 0.9688±0.0002 | **(1)** | 0.6538±0.0003 | **(1)** | 0.7227±0.0012 | **(1)** | 0.7905±0.0026 | **(1)** |
| (PPR) | 5 | **0.5417**±0.0006 | 2E0 | 0.9692±0.0007 | 2E0 | 0.6518±0.0002 | 2E0 | 0.7238±0.0007 | 2E0 | 0.8005±0.0040 | 2E0 |
| +Pooling | 3/5 | 0.5395±0.0013 | 2E0 | **0.9703**±0.0003 | 2E0 | **0.6564**±0.0004 | 1E0 | **0.7258**±0.0017 | 2E0 | **0.8067**±0.0037 | 1E0 |
| GAT | 3 | 0.5070±0.0032 | 2E1 | OOM | 3E2 | OOM | 2E2 | 0.7201±0.0011 | 1E2 | OOM | 3E1 |
| | 5 | 0.5164±0.0033 | 2E2 | OOM | 2E3 | OOM | 2E3 | OOM | 3E3 | OOM | 2E3 |
| GAT-SAINT | 3 | 0.5225±0.0053 | 2E1 | 0.9671±0.0003 | 3E2 | 0.6459±0.0002 | 2E2 | 0.6977±0.0003 | 1E2 | 0.8027±0.0028 | 3E1 |
| | 5 | 0.5153±0.0034 | 2E2 | 0.9651±0.0024 | 2E3 | 0.6478±0.0012 | 2E3 | 0.6954±0.0013 | 3E3 | 0.7990±0.0072 | 2E3 |
| SHADOW-GAT | 3 | 0.5349±0.0023 | **(1)** | 0.9707±0.0004 | **(1)** | **0.6575**±0.0004 | **(1)** | 0.7235±0.0020 | **(1)** | 0.8006±0.0014 | **(1)** |
| (PPR) | 5 | 0.5352±0.0028 | 2E0 | **0.9713**±0.0004 | 2E0 | 0.6559±0.0002 | 2E0 | **0.7274**±0.0022 | 2E0 | 0.8071±0.0004 | 2E0 |
| +Pooling | 3/5 | **0.5364**±0.0026 | 1E0 | 0.9710±0.0004 | 2E0 | 0.6566±0.0005 | 1E0 | 0.7265±0.0028 | 2E0 | **0.8142**±0.0031 | 1E0 |

**Comparison with baselines.** Table 1 shows the performance comparison of SHADOW-GNN with the normal GNNs. All models on all datasets have uniform hidden dimension of 256. We define the metric "inference cost" as the average amount of computation to generate prediction for one test node. Inference cost is a measure of computation complexity (see Appendix B for the calculation) and is independent of hardware / implementation factors such as parallelization strategy, batch processing, *etc*. For cost of SHADOW-GNN, we do not include the overhead on computing EXTRACT, since it is hard to calculate such cost analytically. Empirically, subgraph extraction is much cheaper than model computation (see Figure 8, 9 for time evaluation on CPU and GPU). During training, we apply DropEdge [39] to both the baseline and SHADOW models. DropEdge helps improve the baseline accuracy by alleviating oversmoothing, and benefits SHADOW-GNN due to its regularization effects. See Appendix F.2 for results on other architectures including GIN and JK-Net.

ACCURACY We aim at answering the following questions: 1. Can we improve accuracy by decoupling a baseline model? 2. What architecture components can we tune to improve accuracy of a decoupled model? 3. What EXTRACT can we tune to improve the accuracy of a decoupled model? To answer Q1, we fix the backbone architecture and remove pooling. Then we inspect "3-layer normal GNN *vs.* 3-layer SHADOW-GNN" and "5-layer normal GNN *vs.* 5-layer SHADOW-GNN". Clearly, SHADOW-GNNs (with scope size no more than 200) in general achieve significantly higher accuracy than the normal GNNs. This indicates that a shallow neighborhood contains sufficient information,

and customizing the scope can benefit accuracy even without architecture changes (from Figure 2, a depth-3 $\mathcal{G}_{[v]}$ differs significantly from the 3-hop neighborhood). To answer Q1, we focus on the PPR EXTRACT and thus compare the rows in blue background. We use the 3-layer SHADOW-GNN without pooling as the baseline and analyze the effects of 1. increasing the GNN depth without expanding scope, and 2. adding subgraph pooling. Comparing among the rows in light blue background, we observe that in many cases, simply increasing the depth from 3 to 5 leads to significant accuracy gain. Comparing the ligh blue rows with the dark blue rows, we observe that sometimes pooling can further improve the accuracy of a SHADOW-GNN. In conclusion, both types of architecture tuning are effective ways of optimizing a SHADOW-GNN. Finally, to answer Q3, we compare the light blue rows with the light yellow rows. In general, PPR EXTRACT leads to higher accuracy than 2-hop EXTRACT, demonstrating the importance of designing a good EXTRACT.

INFERENCE COST    Inference cost of SHADOW-GNN is orders of magnitude lower than the normal GNNs (a 5-layer SHADOW-GNN is still much cheaper than a 3-layer normal GNN). The high cost of the baselines is due to the "neighborhood explosion" with respect to more layers. SHADOW-GNN is efficient and scalable as the cost only grows linearly with the model depth. Note that GraphSAINT only improves efficiency *during training* since its inference operates on the full $L$-hop neighborhood.

**Scaling to 100 million nodes.**    We further scale SHADOW-GNN to `ogbn-papers100M`, one of the largest public dataset. Even through the full graph size is at least two orders of magnitude larger than the graphs in Table 1, the localized scope of SHADOW-GNN barely needs to increase. Since SHADOW-GNN performs

Table 2: Leaderboard comparison on `papers100M`

| Method | Test accuracy | Val accuracy | Neigh size |
|---|---|---|---|
| GraphSAGE+incep | $0.6706\pm0.0017$ | $0.7032\pm0.0011$ | 4E5 |
| SIGN-XL | $0.6606\pm0.0019$ | $0.6984\pm0.0006$ | > 4E5 |
| SGC | $0.6329\pm0.0019$ | $0.6648\pm0.0020$ | > 4E5 |
| SHADOW-GAT$_{200}$ | $0.6681\pm0.0016$ | $0.7019\pm0.0011$ | 2E2 |
| SHADOW-GAT$_{400}$ | $\mathbf{0.6708}\pm0.0017$ | $\mathbf{0.7073}\pm0.0011$ | 3E2 |

minibatch computation, a low-end GPU with limited memory capacity can compute SHADOW-GNN on `ogbn-papers100M` efficiently. We show in Appendix F.1 that we can train and inference our model with as little as 4GB GPU memory consumption. This is infeasible using normal GNNs. Table 2 summarizes our comparison with the top leaderboard methods [45, 11, 47]. We only include those methods that do not use node labels as the model input (*i.e.*, the most standard setup). We achieve at least 3 orders of magnitude reduction in neighborhood size without sacrificing accuracy. For SIGN-XL and SGC, their neighborhood is too large to count the exact size. Also, their preprocessing consumes $5\times$ more CPU memory than SHADOW-GNN (Appendix F.1).

**Extending to link-level task.**    We further show that SHADOW-GNN is general and can be extended to the link prediction task. There are two settings of `ogbl-collab`. We follow the one where validation edges cannot be used in training updates. This is the setting which most leaderboard methods follow. Table 3 shows the comparison with the top GNN models

Table 3: Leaderboard comparison on `collab`

| Method | Test Hits@50 | Val Hits@50 |
|---|---|---|
| SEAL | $0.5371\pm0.0047$ | $0.6389\pm0.0049$ |
| DeeperGCN | $0.5273\pm0.0047$ | $0.6187\pm0.0045$ |
| LRGA+GCN | $0.5221\pm0.0072$ | $0.6088\pm0.0059$ |
| SHADOW-SAGE | $\mathbf{0.5492}\pm0.0022$ | $\mathbf{0.6524}\pm0.0017$ |

under the same setting. SHADOW-SAGE outperforms the *rank-1* model with significant margin.

**Oversmoothing.**    To validate Theorem 3.2, we pick SGC as the backbone architecture. SGC with power $L$ is equivalent to $L$-layer GCN without activation. Performance comparison between SGC and SHADOW-SGC thus reveals the effect of oversmoothing without introducing other factors due to optimizing deep neural networks (e.g., vanishing gradients). In Figure 3, we vary the power of SGC and SHADOW-SGC from 1 to 40 (see Appendix E.5 for details). While SGC gradually collapses local information into global "white noise", accuracy of SHADOW-SGC does not degrade. This validates our theory that extracting local subgraphs prevents oversmoothing.

## 6    Conclusion

We have presented a design principle to decouple the depth and scope of GNNs. Applying such a principle on various GNN architectures simultaneously improves expressivity and computation scalability of the corresponding models. We have presented thorough theoretical analysis on expressivity from three different perspectives, and also rich design components (*e.g.*, subgraph extraction functions, architecture extensions) to implement such design principle. Experiments show significant performance improvement over a wide range of graphs, GNN architectures and learning tasks.

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
