# A Proofs

## A.1 Proof on SHADOW-GCN Expressivity

*Proof of Proposition 3.1.* The GCN model performs symmetric normalization on the adjacency matrix. SHADOW-GCN follows the same way to normalize the subgraph adjacency matrix as:

$$\widetilde{\boldsymbol{A}}_{[v]} = \left(\boldsymbol{D}_{[v]} + \boldsymbol{I}_{[v]}\right)^{-\frac{1}{2}} \cdot \left(\boldsymbol{A}_{[v]} + \boldsymbol{I}_{[v]}\right) \cdot \left(\boldsymbol{D}_{[v]} + \boldsymbol{I}_{[v]}\right)^{-\frac{1}{2}} \tag{3}$$

where $\boldsymbol{A}_{[v]} \in \mathbb{R}^{N \times N}$ is the binary adjacency matrix for $\mathcal{G}_{[v]}$.

$\widetilde{\boldsymbol{A}}_{[v]}$ is a real symmetric matrix and has the largest eigenvalue of 1. Since EXTRACT ensures the subgraph $\mathcal{G}_{[v]}$ is connected, so the multiplicity of the largest eigenvalue is 1. By Theorem 1 of [17], we can bound the eigenvalues $\lambda_i$ by $1 = \lambda_1 > \lambda_2 \geq \ldots \geq \lambda_N > -1$.

Performing eigen-decomposition on $\widetilde{\boldsymbol{A}}_{[v]}$, we have

$$\widetilde{\boldsymbol{A}}_{[v]} = \boldsymbol{E}_{[v]} \boldsymbol{\Lambda} \boldsymbol{E}_{[v]}^{-1} = \boldsymbol{E}_{[v]} \boldsymbol{\Lambda} \boldsymbol{E}_{[v]}^{\mathsf{T}} \tag{4}$$

where $\boldsymbol{\Lambda}$ is a diagonal matrix $\Lambda_{i,i} = \lambda_i$ and matrix $\boldsymbol{E}_{[v]}$ consists of all the normalized eigenvectors. We have:

$$\widetilde{\boldsymbol{A}}_{[v]}^{L} = \boldsymbol{E}_{[v]} \boldsymbol{\Lambda}^{L} \boldsymbol{E}_{[v]}^{\mathsf{T}} \tag{5}$$

Since $|\lambda_i| < 1$ when $i \neq 1$, we have $\lim_{L \to \infty} \widetilde{\boldsymbol{A}}_{[v]}^{L} = \boldsymbol{e}_{[v]} \boldsymbol{e}_{[v]}^{\mathsf{T}}$, where $\boldsymbol{e}_v$ is the eigenvector corresponding to $\lambda_1$. It is easy to see that $\left[\boldsymbol{e}_{[v]}\right]_u \propto \sqrt{\delta_{[v]}(u)}$ [17]. After normalization, $\left[\boldsymbol{e}_{[v]}\right]_u = \sqrt{\frac{\delta_{[v]}(u)}{\sum_{w \in \mathcal{V}_{[v]}} \delta_{[v]}(w)}}$.

It directly follows that $\boldsymbol{m}_{[v]} = \left[\boldsymbol{e}_{[v]}\right]_v \cdot \left(\boldsymbol{e}_{[v]}^{\mathsf{T}} \boldsymbol{X}_{[v]}\right)$, with value of $\boldsymbol{e}_{[v]}$ defined above. □

*Proof of Theorem 3.2.* We first prove the case of $\overline{\boldsymbol{m}}_{[v]} = \boldsymbol{m}_{[v]}$. *i.e.*, $\phi_{\mathcal{G}}(v) = 1$.

According to Proposition 3.1, the aggregation for each target node equals $\boldsymbol{m}_{[v]} = \left[\boldsymbol{e}_{[v]}\right]_v \boldsymbol{e}_{[v]}^{\mathsf{T}} \boldsymbol{X}_{[v]}$. Let $N = |\mathcal{V}|$. Let $\bar{\bar{\boldsymbol{e}}}_{[v]} \in \mathbb{R}^{N \times 1}$ be a "padded" vector from $\boldsymbol{e}_{[v]}$, such that the $u^{\text{th}}$ element is $\left[\boldsymbol{e}_{[v]}\right]_u$ if $u \in \mathcal{V}_{[v]}$, and 0, otherwise. Therefore, $\boldsymbol{m}_{[v]} = \left[\boldsymbol{e}_{[v]}\right]_v \bar{\bar{\boldsymbol{e}}}_{[v]}^{\mathsf{T}} \boldsymbol{X}$. Then, the difference in aggregation of two nodes $u$ and $v$ is given by

$$\boldsymbol{m}_{[v]} - \boldsymbol{m}_{[u]} = \left[\boldsymbol{e}_{[v]}\right]_v \bar{\bar{\boldsymbol{e}}}_{[v]}^{\mathsf{T}} \boldsymbol{X} - \left[\boldsymbol{e}_{[u]}\right]_u \bar{\bar{\boldsymbol{e}}}_{[u]}^{\mathsf{T}} \boldsymbol{X} \tag{6}$$

$$= \boldsymbol{\epsilon}^{\mathsf{T}} \boldsymbol{X}, \tag{7}$$

where $\boldsymbol{\epsilon} = \left[\boldsymbol{e}_{[v]}\right]_v \cdot \bar{\bar{\boldsymbol{e}}}_{[v]}^{\mathsf{T}} - \left[\boldsymbol{e}_{[u]}\right]_u \cdot \bar{\bar{\boldsymbol{e}}}_{[u]}^{\mathsf{T}}$.

When two nodes $u$ and $v$ have identical neighborhood as $\mathcal{V}_{[u]} = \mathcal{V}_{[v]}$, then the aggregation vectors are identical as $\boldsymbol{\epsilon} = \boldsymbol{0}$. However, when two nodes have different neighborhoods, we claim that they almost surely have different aggregations. Let us assume the contrary, *i.e.*, for some $v$ and $u$ with $\mathcal{V}_{[v]} \neq \mathcal{V}_{[u]}$, their aggregations are the same: $\boldsymbol{m}_{[v]} = \boldsymbol{m}_{[u]}$. Then we must have $\boldsymbol{\epsilon}^{\mathsf{T}} \boldsymbol{X} = \boldsymbol{0}$.

Note that, given $\mathcal{G}$, EXTRACT and some continuous distribution to generate $\boldsymbol{X}$, there are only *finite* values for $\boldsymbol{\epsilon}$. The reasons are that 1. $\mathcal{G}$ is finite due to which there are only finite possible subgraphs and, 2. even though $\boldsymbol{X}$ can take infinitely many values, $\boldsymbol{\epsilon}$ does not depend on $\boldsymbol{X}$. Each of such $\boldsymbol{\epsilon} \neq \boldsymbol{0}$ defines a hyperplane in $\mathbb{R}^N$ by $\boldsymbol{\epsilon} \cdot \boldsymbol{x} = \boldsymbol{0}$ (where $\boldsymbol{x} \in \mathbb{R}^N$). Let $\mathcal{H}$ be the finite set of all such hyperplanes.

In other words, $\forall i$, $\boldsymbol{X}_{:,i}$ must fall on one of the hyperplanes in $\mathcal{H}$. However, since $\boldsymbol{X}$ is generated from a continuous distribution in $\mathbb{R}^{N \times f}$, $\boldsymbol{X}_{:,i}$ almost surely does not fall on any of the hyperplanes in $\mathcal{H}$. Therefore, for any $v$ and $u$ such that $\mathcal{V}_{[v]} \neq \mathcal{V}_{[u]}$, we have $\boldsymbol{m}_{[v]} \neq \boldsymbol{m}_{[u]}$ almost surely.

For a more general $\phi_{\mathcal{G}}(v)$ applied on $\boldsymbol{m}_{[v]}$, since $\phi_{\mathcal{G}}$ does not depend on $\boldsymbol{X}$, the proof follows exactly the same steps as above.

□

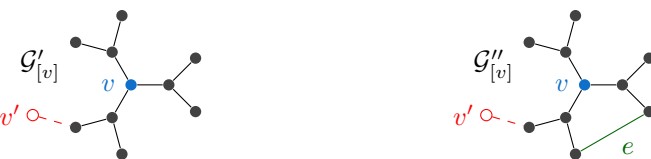

Figure 4: Example $\mathcal{G}'_{[v]}$ and $\mathcal{G}''_{[v]}$ as described in Section 3.2

*Proof of Corollary 3.2.1.* Note that any subgraph contains the target node itself. *i.e.*, $\forall u \in \mathcal{V}$, $u \in \mathcal{V}_{[u]}$. Therefore, for any node $v$, there are at most $n-1$ other nodes in $\mathcal{V}$ with the same neighborhood as $\mathcal{V}_{[v]}$. Such $n-1$ possible nodes are exactly those in $\mathcal{V}_{[v]}$.

By Theorem 3.2, $\forall v \in \mathcal{V}$, there are at most $n-1$ other nodes in $\mathcal{V}$ having the same aggregation as $\boldsymbol{m}_{[v]}$. Equivalently, total number of possible aggregations is at least $\lceil |\mathcal{V}|/n \rceil$. □

*Proof of Corollary 3.2.2.* By definition of EXTRACT$_2$, any pair of nodes has non-identical neighborhood. By Theorem 3.2, any pair of nodes have non-identical aggregation. Equivalently, all nodes have different aggregation and $|\mathcal{M}| = |\mathcal{V}|$. □

## A.2 Proof on SHADOW-SAGE Expressivity

In Section 3.2, we have already shown why the normal GraphSAGE model cannot learn the $\tau$ function. Here we illustrate the idea with an example in Figure 4. The neighborhood $\mathcal{G}'_{[v]}$ (or, $\mathcal{G}''_{[v]}$) is denoted by the solid lines as the edge set $\mathcal{E}'_{[v]}$ (or, $\mathcal{E}''_{[v]}$) and solid nodes as the node set $\mathcal{V}'_{[v]}$ (or, $\mathcal{V}''_{[v]}$). The red dashed edge and the red node $v'$ is in the full graph $\mathcal{G}$ but outside $\mathcal{G}'_{[v]}$ or $\mathcal{G}''_{[v]}$. Due to such $v'$, candidate GraphSAGE models approximating $\tau$ cannot have message passing for more than 2 times. Further, $\mathcal{G}''_{[v]}$ differs from $\mathcal{G}'_{[v]}$ by the green edge $e$ connecting two 2-hop neighbors of the target $v$. The influence of $e$ cannot be propagated to target $v$ by only two message passings. Thus, as discussed in Section 3.2, there is no way for GraphSAGE to learn the $\tau$ function.

*Proof of Theorem 3.3.* SHADOW-SAGE follows the GraphSAGE way of normalization on the adjacency matrix. Thus, each row of $\widehat{\boldsymbol{A}}_{[v]}$ sums up to 1. Such normalization enables us to view $\widehat{\boldsymbol{A}}_{[v]}$ as the transition matrix of a Markov chain. The coefficients of the linear function $\tau$ (*i.e.*, $\delta_{[v]}(\cdot)$) equal the limiting distribution of such a Markov chain. Therefore, we can use the convergence theorem of Markov chain to characterize the convergence of $\widehat{\boldsymbol{A}}_{[v]}^{L'}$ towards $\tau$'s coefficients. We can also use the mixing time of the Markov chain to derive the error bound of shaDow-SAGE. Our proof is built on the existing theoretical results in [27].

We rewrite $\tau$ as $\tau = C \cdot \boldsymbol{\delta X}$, where $\boldsymbol{\delta}$ is a length-$|\mathcal{V}|$ vector with $|\mathcal{V}_{[v]}|$ non-zero elements – each non-zero corresponds to $\delta_{[v]}(u)$ of the neighborhood $\mathcal{G}_{[v]}$. Further, denote $\widehat{\boldsymbol{\delta}}$ as the normalized $\boldsymbol{\delta}$ vector. *i.e.*, each non-zero element of $\widehat{\boldsymbol{\delta}}$ equals $\frac{\delta_{[v]}(u)}{\sum_{w \in \mathcal{V}_{[v]}} \delta_{[v]}(w)}$. So $\tau = C' \cdot \widehat{\boldsymbol{\delta}} \boldsymbol{X}$ with some scaling factor $C'$. For ease of discussion, we ignore $C'$, as any scaling factor can be easily expressed by the model weight parameters.

By Section 3.2, SHADOW-SAGE can express $\left[\widehat{\boldsymbol{A}}_{[v]}^{L'}\boldsymbol{X}\right]_{v,:} = \left[\widehat{\boldsymbol{A}}_{[v]}^{L'}\right]_{v,:}\boldsymbol{X}$. So now we need to show how $\left[\widehat{\boldsymbol{A}}_{[v]}^{L'}\right]_{v,:}$ converges to $\widehat{\boldsymbol{\delta}}$ when $L' \to \infty$. We establish the following correspondence:

$\widehat{\boldsymbol{A}}_{[v]}$ **as the Markov transition matrix.** This can be easily seen since each row of $\widehat{\boldsymbol{A}}_{[v]}$ sums up to 1. Further, if we add self-loop in $\mathcal{V}_{[v]}$ and we consider $\mathcal{G}_{[v]}$ as undirected and connected (as can be guaranteed by EXTRACT), then $\widehat{\boldsymbol{A}}_{[v]}$ is *irreducible*, *aperiodic* and *reversible*. Irreducibility is guaranteed by $\mathcal{G}_{[v]}$ being connected. Aperiodicity is guaranteed by self-loops. For reversibility, we

can prove it by the stationary distribution of $\widehat{\boldsymbol{A}}_{[v]}$. As shown by the next paragraph, $\widehat{\boldsymbol{\delta}}$ is the stationary distribution of $\widehat{\boldsymbol{A}}_{[v]}$. So we need to show that

$$\left[\widehat{\boldsymbol{\delta}}\right]_u \left[\widehat{\boldsymbol{A}}_{[v]}\right]_{u,w} = \left[\widehat{\boldsymbol{\delta}}\right]_w \left[\widehat{\boldsymbol{A}}_{[v]}\right]_{w,u} \tag{8}$$

Consider two cases. If $(u,w) \notin \mathcal{E}_{[v]}$, then $(w,u) \notin \mathcal{E}_{[v]}$ and both sides of Equation 8 equal 0. If $(u,w) \in \mathcal{E}_{[v]}$, then $\left[\widehat{\boldsymbol{\delta}}\right]_u = \frac{\delta_{[v]}(u)}{\sum_{x \in \mathcal{V}_{[v]}} \delta_{[v]}(x)}$ and $\left[\widehat{\boldsymbol{A}}_{[v]}\right]_{u,w} = \frac{1}{\delta_{[v]}(u)}$. So both sides of Equation 8 equal $\frac{1}{\sum_{x \in \mathcal{V}_{[v]}} \delta_{[v]}(x)}$. Thus, Equation 8 holds and $\widehat{\boldsymbol{A}}_{[v]}$ is reversible.

**$\widehat{\boldsymbol{\delta}}$ as the limiting distribution.** By definition, the stationary distribution $\boldsymbol{\pi}$ satisfies

$$\boldsymbol{\pi} = \boldsymbol{\pi}\widehat{\boldsymbol{A}}_{[v]} \tag{9}$$

It is easy to see that setting $\boldsymbol{\pi} = \widehat{\boldsymbol{\delta}}^{\mathsf{T}}$ can satisfy Equation 9. So $\widehat{\boldsymbol{\delta}}$ is the stationary distribution. For irreducible and aperiodic Markov chain, the stationary distribution is also the *limiting distribution*, and thus

$$\lim_{L' \to \infty} \left[\widehat{\boldsymbol{A}}_{[v]}^{L'}\right]_{v,:} = \widehat{\boldsymbol{\delta}} \tag{10}$$

So far, we can already show that when the SHADOW-SAGE model is deep enough (*i.e.*, with large $L'$), the model output approximates $\tau$:

$$\lim_{L' \to \infty} \left[\widehat{\boldsymbol{A}}_{[v]}^{L'}\boldsymbol{X}\right]_{v,:} = \tau \tag{11}$$

**Desired model depth as the mixing time.** Next, we want to see if we want the SHADOW-SAGE model to reach a given error bound, how many layers $L'$ are required. Firstly, directly applying Theorem 4.9 of [27], we know that the error of SHADOW-SAGE approximating $\tau$ reduces *exponentially* with the model depth $L'$. Then From Equation 4.36 of [27], we can directly relate the mixing time of Markov chain with the required shaDow-SAGE depth to reach any $\epsilon$ error. Finally, by Theorem 12.3 of [27], the mixing time can be bounded by the "absolute spectral gap" of the transition matrix $\widehat{\boldsymbol{A}}_{[v]}$. Note that Theorem 12.3 applies when the Markov chain is reversible – a condition satisfied by $\widehat{\boldsymbol{A}}_{[v]}$ as we have already discussed. The absolute spectral gap is calculated from the eigenvalues of the transition matrix $\widehat{\boldsymbol{A}}_{[v]}$.

In summary, SHADOW-SAGE can approximate $\tau$ with error decaying exponentially with depth $L'$. $\qquad \square$

### A.3 Proof on SHADOW-GIN Expressivity

*Proof of Theorem 3.4.* We consider GNNs whose layer function is defined by Equation 2. Define $\mathcal{G}_{[v]}^L$ as the subgraph *induced* from all of $v$'s $\ell$-hop neighbors in $\mathcal{G}$, where $1 \le \ell \le L$.

First, we show that a SHADOW-GNN following Equation 2 is at least as discriminative as the 1-dimensional Weisfeiler-Lehman test (1-WL). We first prove that given any $\mathcal{G}_{[v]}^L$, an $L'$-layer GNN can have exactly the same output as an $L$-layer GNN, where $L' > L$. We note that for the target node $v$, the only difference between these two architectures is that an $L$-layer GNN exactly performs $L$ message passing iterations to propagate node information from at most $L$ hops away, while an $L'$-layer GNN has $L' - L$ more message passing iterations before performing the rest of $L$ message passings. Thanks to the universal approximation theorem [15], we can let the MLPs implementing $f_1$ and $f_2$ learn the following GNN layer function:

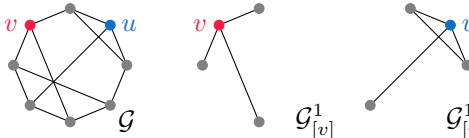

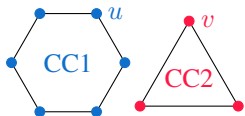

Figure 5: Example 3-regular graph and the 1-hop subgraph of the target nodes

Figure 6: Example 2-regular graph with two connected components (CC)

$$h_v^{(\ell)} = f_1^{(\ell)} \left( h_v^{(\ell-1)}, \sum_{u \in \mathcal{N}_v} f_2^{(\ell)} \left( h_v^{(\ell-1)}, h_u^{(\ell-1)} \right) \right)$$
$$= h_v^{(\ell-1)}, \qquad \forall 1 \le \ell \le L' - L$$

This means $h_v^{(0)} = h_v^{(L'-L)}$. Then the $L'$-layer GNN will have the same output as the $L$-layer GNN. According to [49], the normal GNN (*i.e.*, $L$-layer on $\mathcal{G}_{[v]}^L$) following Equation 2 is as discriminative as 1-WL. Thus, SHADOW-GNN (*i.e.*, $L'$-layer on $\mathcal{G}_{[v]}^L$) following Equation 2 is at least as discriminative as 1-WL.

Next, we show that there exists a graph where SHADOW-GNN can discriminate topologically different nodes, while 1-WL cannot. The example graph mentioned in Section 3.3 is one such graph. We duplicate the graph here in Figure 5. $\mathcal{G}$ is connected and is 3-regular. The nodes $u$ and $v$ marked in red and blue have different topological roles, and thus an ideal model / algorithm should assign different labels to them. Suppose all nodes in $\mathcal{G}$ have identical features. For 1-WL, it will assign the same label to all nodes (including $u$ and $v$) no matter how many iterations it runs. For SHADOW-GNN, if we let EXTRACT return $\mathcal{G}_{[v]}^1$ and $\mathcal{G}_{[u]}^1$ (*i.e.*, 1-hop based EXTRACT), and set $L' > 1$, then our model can assign different labels to $u$ and $v$. To see why, note that $\mathcal{G}_{[u]}^1$ and $\mathcal{G}_{[v]}^1$ are non-isomorphic, and more importantly, *non-regular*. So if we run the "SHADOW" version of 1-WL (*i.e.*, 1-WL on $\mathcal{G}_{[v]}^1$ and $\mathcal{G}_{[u]}^1$ rather than on $\mathcal{G}$), the nodes $u$ and $v$ will be assigned to different labels after two iterations. Equivalently, SHADOW-GNN can discriminate $u$ and $v$ with at least 2 layers.

We can further generalize to construct more such example graphs (although such generalization is not required by the proof). The guideline we follow is that, the full graph $\mathcal{G}$ should be regular. Yet the subgraphs around topologically different nodes (*e.g.*, $\mathcal{G}_{[v]}^k$) should be non-isomorphic and non-regular.

The graph in Figure 6 is another example 2-regular graph, where nodes $u$ and $v$ can only be differentiated by decoupling the GIN.

Finally, combining the above two parts, SHADOW-GNN following Equation 2 is more discriminative than the 1-dimensional Weisfeiler-Lehman test.

$\square$

# B  Inference Complexity Calculation

Here we describe the equations to compute the "inference cost" of Table 1. Recall that inference cost is a measure of computation complexity to generate node embeddings for a given GNN architecture.

The numbers in Table 1 shows on average, how many arithmetic operations is required to generate the embedding for each node. For a GNN layer $\ell$, denote the number of input nodes as $n^{(\ell-1)}$ and the number of output nodes as $n^{(\ell)}$. Denote the number of edges connecting the input and output nodes as $m^{(\ell)}$. Recall that we use $d^{(\ell)}$ to denote the number of channels, or, the hidden dimension.

In the following, we ignore the computation cost of non-linear activation, batch-normalization and applying bias, since their complexity is negligible compared with the other operations.

For the *GCN* architecture, each layer mainly performs two operations: aggregation of the neighbor features and linear transformation by the layer weights. So the number of multiplication-addition (MAC) operations of a layer-$\ell$ equals:

$$P_{\text{GCN}}^{(\ell)} = m^{(\ell)} d^{(\ell-1)} + n^{(\ell)} d^{(\ell-1)} d^{(\ell)} \tag{12}$$

Similarly, for the *GraphSAGE* architecture, the number of MAC operations equals:

$$P_{\text{SAGE}}^{(\ell)} = m^{(\ell)} d^{(\ell-1)} + 2 \cdot n^{(\ell)} d^{(\ell-1)} d^{(\ell)} \tag{13}$$

where the 2 factor is due to the weight transformation on the self-features.

For *GAT*, suppose the number of attention heads is $t$. Then the layer contains $t$ weight matrices $\boldsymbol{W}^i$, each of shape $d^{(\ell-1)} \times \frac{d^{(\ell)}}{t}$. We first transform each of the $n^{(\ell-1)}$ nodes by $\boldsymbol{W}^i$. Then for each edge $(u, v)$ connecting the layer input $u$ to the layer output $v$, we obtain its edge weight (*i.e.*, a scalar) by computing the dot product between $u$'s, $v$'s transformed feature vectors and the model attention weight vector. After obtaining the edge weight, the remaining computation is to aggregate the $n^{(\ell-1)}$ features into the $n^{(\ell)}$ nodes. The final output is obtained by concatenating the features of different heads. The number of MAC operations equals:

$$
\begin{aligned}
P_{\text{GAT}}^{(\ell)} &= t \cdot n^{(\ell-1)} d^{(\ell-1)} \frac{d^{(\ell)}}{t} + 2t \cdot m^{(\ell)} \frac{d^{(\ell)}}{t} + m^{(\ell)} d^{(\ell)} \\
&= 3 m^{(\ell)} d^{(\ell)} + n^{(\ell-1)} d^{(\ell-1)} d^{(\ell)}
\end{aligned} \tag{14}
$$

On the same graph, GCN is less expensive than GraphSAGE. GraphSAGE is in general less expensive than GAT, due to $n^{(\ell-1)} > n^{(\ell)}$ (on the other hand, note that $n^{(\ell-1)} = n^{(\ell)}$ for any SHADOW-GNN). In addition, for all architectures, $P_*^{(\ell)}$ grows proportionally with $n^{(\ell)}$ and $m^{(\ell)}$. For the normal GNN architecture, since we are using the full $\ell$-hop neighborhood for each node, the value of $n^{(\ell)}$ and $m^{(\ell)}$ may grow exponentially with $\ell$. This is the "neighbor explosion" phenonemon and is the root cause of the high inference cost of the baseline in Table 1.

For SHADOW-GNN, suppose the subgraph contains $n$ nodes and $m$ edges. Then $n^{(\ell)} = n$ and $m^{(\ell)} = m$. The inference cost of any SHADOW-GNN is ensured to only grow linearly with the depth of the GNN.

**Remark**    The above calculation on normal GNNs is under the realistic setting of minibatch inference. By "realistic setting", we mean 1. the graph size is often gigantic (of similar scale as `ogbn-papers100M`); 2. the total number of target nodes can be much, much smaller than the full graph size (*e.g.*, $< 1\%$ of all nodes are target nodes, as in `ogbn-papers100M`), and 3. we may only want to generate embedding for a small subset of the target nodes at a time. As a result, the above calculated computation complexity reflects what we can achieve in practice, under minibatch computation.

On the other hand, most of the benchmark graphs evaluated in the paper may not reflect the realistic setting. For example, `ogbn-arxiv` is downscaled $657\times$ from `ogbn-papers100M`. Consequently, the full graph size is small and the number of target nodes is comparable to the full graph size. Such a "benchmark setting" enables full-batch inference: *e.g.*, for GCN, one multiplication on the full graph adjacency matrix (*i.e.*, $\widetilde{\boldsymbol{A}} \cdot \boldsymbol{H} \cdot \boldsymbol{W}$) generates the next layer hidden features for all nodes at once.

While full-batch computation leads to significantly lower computation complexity than Equations 12, 13 and 14, it has strict constraints on the GPU memory size and the graph size – the full adjacency matrix and the various feature data of *all* nodes need to completely fit into the GPU memory. For example, according to the official OGB guideline [16], full-batch training of `ogbn-products` on a 3-layer GNN requires 48GB of GPU memory. On one hand, only the most powerful NVIDIA GPUs (*e.g.*, RTX A6000) have memory as large as 48GB. On the other hand, `ogbn-products` is still $45\times$ smaller than `ogbn-papers100M`. Therefore, full batch computation is not feasible in practice.

In summary, the inference complexity shown in Table 1 based on Equations 12, 13 and 14 reflects the feasibility of deploying the GNN models in *real-life applications*.

## C Designing Subgraph Extraction Algorithms

We illustrate how we can follow the three main approaches (*i.e.*, heuristic based, model based and learning based. See Section 3.4) to design various subgraph extraction functions. Note that the $L$-hop and PPR based EXTRACT are only example algorithms to implement the principle of "decoupling the depth and scope".

**Heuristic based.** As discussed in Section 3.4, we can design EXTRACT by selecting appropriate graph metrics. We have shown some examples in Section 3.4. Here we provide detailed description on the two EXTRACT used in our experiments.

$L$-HOP EXTRACT. Starting from the target node $v$, the algorithm traverses up to $L$ hops. At a hop-$\ell$ node $u$, the sampler will add to its hop-$(\ell + 1)$ node set either all neighbors of $u$, or $b$ randomly selected neighbors of $u$. The subgraph $\mathcal{G}_{[v]}$ is induced from all the nodes selected by the EXTRACT. Here depth $L$ and budget $b$ are the hyperparameters.

PPR EXTRACT. Given a target $v$, our PPR EXTRACT proceeds as follows: 1. Compute the approximate PPR vector $\boldsymbol{\pi}_v \in \mathbb{R}^{|\mathcal{V}|}$ for $v$. 2. Select the neighborhood $\mathcal{V}_{[v]}$ such that for $u \in \mathcal{V}_{[v]}$, the PPR score $[\boldsymbol{\pi}_v]_u$ is large. 3. Construct the induced subgraph from $\mathcal{V}_{[v]}$. For step 1, even though vector $\boldsymbol{\pi}_v$ is of length-$|\mathcal{V}|$, most of the PPR values are close to zero. So we can use a fast algorithm [3] to compute the approximate PPR score by traversing only the local region around $v$. Throughout the $\boldsymbol{\pi}_v$ computation, most of its entries remain as the initial value of 0. Therefore, the PPR EXTRACT is scalable and efficient w.r.t. both time and space complexity. For step 2, we can either select $\mathcal{V}_{[v]}$ based on top-$p$ values in $\boldsymbol{\pi}_v$, or based on some threshold $[\boldsymbol{\pi}_v] > \theta$. Then $p, \theta$ are hyperparameters. For step 3, notice that our PPR sampler only uses PPR scores as a node filtering condition. The original graph structure is still preserved among $\mathcal{V}_{[v]}$, due to the induced subgraph step. In comparison, other related works [23, 6] do not preserve the original graph structure.

We also empirically highlight that the approximate PPR is both scalable and efficient. The number of nodes we need to visit to obtain the approximate $\boldsymbol{\pi}_v$ is much smaller than the graph size $|\mathcal{V}|$. In addition, each visit only involves a scalar update, which is orders of magnitude cheaper than the cost of neighbor propagation in a GNN. We profile the three datasets, Reddit, Yelp and ogbn-products. As shown in Table 4, for each target node on average, the number of nodes touched by the PPR computation is comparable to the full 2-hop neighborhood size. This also indicates that faraway neighbors do not have much topological significance. We further show the empirical execution time measurement on multi-core CPU machines in Figure 8.

Table 4: Average number of nodes touched by the approximate PPR computation

| Dataset | Avg. 2-hop size | Avg. # nodes touched by PPR |
|---|---|---|
| Reddit | 11093 | 27751 |
| Yelp | 2472 | 5575 |
| ogbn-products | 3961 | 5405 |

**Model based.** From Section 1, treating $\mathcal{G}$ as the union of $\mathcal{G}_{[v]}$'s describes a generation process on $\mathcal{G}$. EXTRACT then describes the *reverse* of such a process. This links to the graph generation literature [26]. *e.g.*, forest-fire generation model would correspond to EXTRACT being forest-fire sampler [25]. More specifically, to extract the subgraph around a node $v$, we can imagine a process of adding $v$ into the partial graph $\mathcal{G}'$ consisting of $\mathcal{V} \setminus \{v\}$. Then the nodes selected by EXTRACT would correspond to the nodes touched by such an imaginary process of adding $v$.

**Learning based.** We may treat the design of EXTRACT as part of GNN training. However, due to the combinatorial nature of subgraph extraction, simultaneously learning EXTRACT and the SHADOW-GNN layer parameters is challenging. The learning of EXTRACT can be made possible with appropriate approximation and relaxations. *e.g.*, we may use the techniques proposed in [51] to design a two-phased learning process. In Phase 1, we train a normal $L$-layer GNN, and then use [51] to identify the important neighbors among the full $L$-hop neighborhood. In Phase 2, we use the neighborhood identified in Phase 1 as the subgraph returned by EXTRACT. Then we train an $L'$-layer SHADOW-GNN on top of such neighborhood.

Detailed design and evaluation on the model and learning based approaches are left as future work.

---
**Algorithm 1** Inference algorithm for the general SHADOW-GNN model
---

**Input:** $\mathcal{G}(\mathcal{V}, \mathcal{E}, \boldsymbol{X})$; Target nodes $\mathcal{V}_t$; GNN model; $C$ number of samplers $\{\texttt{EXTRACT}_i\}$;
**Output:** Node embedding matrix $\boldsymbol{Y}_t$ for $\mathcal{V}_t$;
**for** $v \in \mathcal{V}_t$ **do**
    **for** $i = 1$ to $C$ **do**
        Get $\mathcal{G}_{[v],i}$ by $\texttt{EXTRACT}_i$ on $\mathcal{G}$
        Propagate $\mathcal{G}_{[v],i}$ in the $i^{\text{th}}$ branch of $L$-layer GNN
        $\boldsymbol{H}_{[v],i} \leftarrow f_i^L\left(\boldsymbol{X}_{[v],i}, \boldsymbol{A}_{[v],i}\right)$
        $\boldsymbol{y}_{v,i} \leftarrow \texttt{MLP}\left(\texttt{READOUT}\left(\boldsymbol{H}_{[v],i}\right) \big\| \left[\boldsymbol{H}_{[v],i}\right]_{v,:}\right)$
    **end for**
    $\boldsymbol{y}_v \leftarrow \texttt{ENSEMBLE}\left(\{\boldsymbol{y}_{v,i}\}\right)$
**end for**
---

## D  The Complete SHADOW-GNN Framework

Algorithm 1 shows the inference algorithm of SHADOW-GNN after integrating the various architecture extensions discussed in Section 3.5. The $f_i$ function specifies the layer propagation of a given GNN architecture (*e.g.*, GCN, GraphSAGE, *etc.*), and $f_i^L$ is a shorthand for $L$ times iterative layer propagation of the $L$-layer model. The $\texttt{READOUT}(\cdot)$ function performs subgraph pooling and $\texttt{ENSEMBLE}(\cdot)$ performs subgraph ensemble. The implementation of such functions are described in the following subsections.

After the pooling by $\texttt{READOUT}(\cdot)$, we further feed into an MLP the vector summarized from the subgraph and the vector for the target. This way, even if two target nodes $u$ and $v$ have the same neighborhood, we can still differentiate their embeddings based on the vectors $\left[\boldsymbol{H}_{[v]}\right]_{v,:}$ and $\left[\boldsymbol{H}_{[u]}\right]_{u,:}$.

### D.1  Architecture for Subgraph Pooling

The $\texttt{READOUT}(\cdot)$ function in Algorithm 1 can perform subgraph-level operation such as sum pooling, max pooling, mean pooling, sort pooling [57], *etc.* Table 5 summarizes all the pooling operations that we have integrated into the SHADOW-GNN framework, where $\boldsymbol{H}_{[v]}$ is the subgraph embedding matrix as shown in Algorithm 1.

Table 5: $\texttt{READOUT}(\cdot)$ function for different pooling operations.

| Name | $\texttt{READOUT}(\cdot)$ |
|---|---|
| Center | $\left[\boldsymbol{H}_{[v]}\right]_{v,:}$ |
| Sum | $\sum_{u \in \mathcal{V}_{[v]}} \left[\boldsymbol{H}_{[v]}\right]_{u,:}$ |
| Mean | $\frac{1}{|\mathcal{V}_{[v]}|} \sum_{u \in \mathcal{V}_{[v]}} \left[\boldsymbol{H}_{[v]}\right]_{u,:}$ |
| Max | $\max_{u \in \mathcal{V}_{[v]}} \left[\boldsymbol{H}_{[v]}\right]_{u,:}$ |
| Sort | $\texttt{MLP}\left(\left[\boldsymbol{H}_{[v]}\right]_{\left[\arg\text{sort}\left[\boldsymbol{H}_{[v]}\right]_{:,-1}\right]_{:s}}\right)$ |

In particular, sort pooling 1. sorts the last column of the feature matrix, 2. takes the indices of the top $s$ values after sorting ($s$ is a hyperparameter of the sort pooling function), and 3. slice a submatrix of $\boldsymbol{H}_{[v]}$ based on the top-$s$ indices. The input to the $\texttt{MLP}(\cdot)$ (last row of Table 5) is a submatrix of $\boldsymbol{H}_{[v]}$ consisting of $s$ rows, and the output of the $\texttt{MLP}(\cdot)$ is a single vector.

### D.2  Architecture for Subgraph Ensemble

For the ensemble function, we implement the following after applying attention on the outputs of the different model branches:

$$w_i = \text{MLP}\left(\boldsymbol{y}_{v,i}\right) \cdot \boldsymbol{q} \tag{15}$$

$$\boldsymbol{y}_v = \sum_{i=1}^{C} \widetilde{w}_i \cdot \boldsymbol{y}_{v,i} \tag{16}$$

where $\boldsymbol{q}$ is a learnable vector; $\widetilde{\boldsymbol{w}}$ is normalized from $\boldsymbol{w}$ by softmax; $\boldsymbol{y}_v$ is the final embedding.

# E    Detailed Experimental Setup

We used the tool "Weight & Biases" [5] for experiment tracking and visualization to develop insights for this paper.

## E.1    Additional Dataset Details

The statistics for the seven benchmark graphs is listed in Table 6. Note that for `Yelp`, each node may have multiple labels, and thus we follow the original paper [55] to report its F1-micro score. For all the other node classification graphs, a node is only associated with a single label, and so we report accuracy. Note that for `Reddit` and `Flickr`, other papers [12, 55] also report F1-micro score. However, since each node only has one label, "F1-micro score" is exactly the same as "accuracy". For the link prediction graph `ogbl-collab`, we use "Hits@50" as the metric.

For `ogbn-papers100M`, only around $1\%$ of the nodes are associated with ground truth labels. The training / validation / test sets are split only among those labelled nodes.

Table 6: Dataset statistics

| Dataset | Setting | Nodes | Edges | Degree | Feature | Classes | Train / Val / Test |
|---|---|---|---|---|---|---|---|
| Flickr | Inductive | 89,250 | 899,756 | 10 | 500 | 7 | 0.50 / 0.25 / 0.25 |
| Reddit | Inductive | 232,965 | 11,606,919 | 50 | 602 | 41 | 0.66 / 0.10 / 0.24 |
| Yelp | Inductive | 716,847 | 6,977,410 | 10 | 300 | 100 | 0.75 / 0.10 / 0.15 |
| ogbn-arxiv | Transductive | 169,343 | 1,166,243 | 7 | 128 | 40 | 0.54 / 0.18 / 0.29 |
| ogbn-products | Transductive | 2,449,029 | 61,859,140 | 25 | 100 | 47 | 0.10 / 0.02 / 0.88 |
| ogbn-papers100M | Transductive | 111,059,956 | 1,615,685,872 | 29 | 128 | 172 | 0.78 / 0.08 / 0.14 |
| ogbl-collab | – | 235,868 | 1,285,465 | 8 | 128 | – | 0.92/0.04/0.04 |

Note that `Reddit` is a pre-exisiting dataset collected by Stanford, available at `http://snap.stanford.edu/graphsage`. Facebook did not directly collect any data from Reddit. None Reddit content is reproduced in this paper.

## E.2    Hardware

We have tested SHADOW-GNN under various hardware configurations, ranging from low-end desktops to high-end servers. We observe that the training and inference of SHADOW-GNN can be easily adapted to the amount of available hardware resources by adjusting the batch size.

In summary, we have used the following three machines to compute SHADOW-GNN.

- MACHINE 1: This is a low-end desktop machine with 4-core Intel Core i7-6700 CPU @3.4GHz, 16GB RAM and one NVIDIA GeForce GTX 1650 GPU of 4GB memory.
- MACHINE 2: This is a low-end server with 28-core Intel Xeon Gold 5120 CPU @2.2GHz, 128GB RAM and one NVIDIA Titan Xp GPU of 12GB memory.
- MACHINE 3: This is another server with 64-core AMD Ryzen Threadripper 3990x CPU @2.9GHz, 256GB RAM and three NVIDIA GeForce RTX3090 GPUs of 24GB memory.

From the GPU perspective, the low-end GTX 1650 GPU can support the SHADOW-GNN computation on all the graphs (including `ogbn-papers100M`). However, the limited RAM size of MACHINE 1 limits its usage to only `Flickr`, `Reddit`, `ogbn-arxiv` and `ogbl-collab`. The other two servers, MACHINE 2 and MACHINE 3, are used to train all of the seven graphs.

Table 7 summarizes our recommended minimum hardware resources to run SHADOW-GNN. Note that regardless of the model, larger graphs *inevitibly requires larger RAM* due to the growth of the raw features (the raw data files of `ogbn-papers100M` already takes 70GB). However, larger graphs *do not correspond to higher GPU requirement* for SHADOW-GNN, since the GPU memory consumption is controlled by the batch size parameter.

See Appendix F.1 for how to control the GPU memory-speed tradeoff by batch size.

Table 7: Recommended minimum hardware resources for SHADOW-GNN

| Dataset | Num. nodes | CPU cores | RAM | GPU memory |
|---|---|---|---|---|
| `ogbn-arxiv` | 0.2M | 4 | 8GB | 4GB |
| `ogbn-products` | 2.4M | 4 | 32GB | 4GB |
| `ogbn-papers100M` | 111.1M | 4 | 128GB | 4GB |

### E.3 Software

The code is written in `Python 3.8.5` (where the sampling part is written with `C++` parallelized by `OpenMP`, and the interface between `C++` and `Python` is via `PyBind11`). We use `PyTorch 1.7.1` on `CUDA 11.1` to train the model on GPU.

### E.4 Hyperparameter Tuning

For all models in Table 1, we set the hidden dimension to be $d^{(\ell)} = 256$. In addition, for GAT and SHADOW-GAT, we set the number of attention heads to be $t = 4$. For all the GIN and SHADOW-GIN experiments, we use a 2-layer MLP (with hidden dimension 256) to perform the injective mapping in each GIN layer. For all the JK-Net and SHADOW-JK experiments, we use the concatenation operation to aggregate the hidden features of each layer in the JK layer. Additional comparison on GIN, JK-Net and their SHADOW versions is shown in Table 12.

All experiments are repeated five times without fixing random seeds.

For all the baseline and SHADOW-GNN experiments, we use Adam optimizer [21]. We perform grid search on the hyperparameter space defined by:

- Activation: $\{\texttt{ReLU}, \texttt{ELU}, \texttt{PRELU}\}$
- Dropout: 0 to 0.5 with stride of 0.05
- DropEdge: 0 to 0.5 with stride of 0.05
- Learning rate: $\{1e-2, 2e-3, 1e-3, 2e-4, 1e-4, 2e-5\}$
- Batch size: $\{16, 32, 64, 128\}$

The `EXTRACT` hyperparameters are tuned as follows.

For the PPR `EXTRACT`, we consider two versions: one based on fixed sampling budget $p$ and the other based on PPR score thresholding $\theta$.

- If with fixed budget, then we disable the $\theta$ thresholding. We tune the budget by $p \in \{150, 175, 200\}$.
- If with thresholding, we set $\theta \in \{0.01, 0.05\}$. We still have an upper bound $p$ on the subgraph size. So if there are $q$ nodes in the neighborhood with PPR score larger than $\theta$, the final subgraph size would be $\max\{p, q\}$. Such an upper bound eliminates the corner cases which may cause hardware inefficiency due to very large $q$. In this case, we set the upper bound $p$ to be either 200 or 500.

For $L$-hop `EXTRACT`, we define the hyperparameter space as:

- Depth $L = 2$
- Budget $b \in \{5, 10, 15, 20\}$

Table 8: Training configuration of SHADOW-GNN for Table 1 (PPR EXTRACT)

| Arch. | Dataset | Layers | Dim. | Pooling | Learning Rate | Batch Size | Dropout | DropEdge | Budget $p$ |
|---|---|---|---|---|---|---|---|---|---|
| SHADOW-GCN | Flickr | 3 | 256 | – | 0.001 | 256 | 0.40 | 0.10 | 200 |
| | | 5 | 256 | – | 0.001 | 256 | 0.40 | 0.10 | 200 |
| | | 5 | 256 | mean | 0.001 | 256 | 0.40 | 0.10 | 200 |
| | Reddit | 3 | 256 | – | 0.0001 | 128 | 0.20 | 0.15 | 200 |
| | | 5 | 256 | – | 0.0001 | 128 | 0.20 | 0.15 | 200 |
| | | 3 | 256 | max | 0.0001 | 128 | 0.20 | 0.15 | 200 |
| | Yelp | 3 | 256 | – | 0.001 | 32 | 0.10 | 0.00 | 100 |
| | | 5 | 256 | – | 0.001 | 32 | 0.10 | 0.00 | 100 |
| | | 5 | 256 | max | 0.001 | 32 | 0.10 | 0.00 | 100 |
| | ogbn-arxiv | 3 | 256 | – | 0.00005 | 32 | 0.20 | 0.10 | 200 |
| | | 5 | 256 | – | 0.00005 | 32 | 0.20 | 0.10 | 200 |
| | | 5 | 256 | max | 0.00002 | 16 | 0.25 | 0.10 | 200 |
| | ogbn-products | 3 | 256 | – | 0.002 | 128 | 0.40 | 0.05 | 150 |
| | | 5 | 256 | – | 0.002 | 128 | 0.40 | 0.05 | 150 |
| | | 5 | 256 | max | 0.002 | 128 | 0.40 | 0.05 | 150 |
| SHADOW-SAGE | Flickr | 3 | 256 | – | 0.0005 | 64 | 0.45 | 0.05 | 200 |
| | | 5 | 256 | – | 0.001 | 128 | 0.45 | 0.00 | 200 |
| | | 5 | 256 | max | 0.001 | 128 | 0.45 | 0.00 | 200 |
| | Reddit | 3 | 256 | – | 0.0001 | 128 | 0.20 | 0.15 | 200 |
| | | 5 | 256 | – | 0.0001 | 128 | 0.20 | 0.15 | 200 |
| | | 5 | 256 | max | 0.0001 | 128 | 0.20 | 0.15 | 200 |
| | Yelp | 3 | 256 | – | 0.0005 | 16 | 0.10 | 0.00 | 100 |
| | | 5 | 256 | – | 0.0005 | 16 | 0.10 | 0.00 | 100 |
| | | 3 | 256 | max | 0.0005 | 16 | 0.10 | 0.00 | 100 |
| | ogbn-arxiv | 3 | 256 | – | 0.00002 | 16 | 0.25 | 0.15 | 200 |
| | | 5 | 256 | – | 0.00002 | 16 | 0.25 | 0.15 | 200 |
| | | 5 | 256 | max | 0.00002 | 16 | 0.25 | 0.15 | 200 |
| | ogbn-products | 3 | 256 | – | 0.002 | 128 | 0.40 | 0.05 | 150 |
| | | 5 | 256 | – | 0.002 | 128 | 0.40 | 0.05 | 150 |
| | | 3 | 256 | max | 0.002 | 128 | 0.40 | 0.15 | 150 |
| SHADOW-GAT | Flickr | 3 | 256 | – | 0.0005 | 64 | 0.45 | 0.00 | 200 |
| | | 5 | 256 | – | 0.0005 | 64 | 0.45 | 0.00 | 200 |
| | | 3 | 256 | mean | 0.001 | 128 | 0.40 | 0.00 | 200 |
| | Reddit | 3 | 256 | – | 0.0001 | 128 | 0.20 | 0.00 | 200 |
| | | 5 | 256 | – | 0.0001 | 128 | 0.20 | 0.00 | 200 |
| | | 5 | 256 | max | 0.0001 | 128 | 0.20 | 0.00 | 200 |
| | Yelp | 3 | 256 | – | 0.0005 | 16 | 0.10 | 0.00 | 100 |
| | | 5 | 256 | – | 0.0005 | 16 | 0.10 | 0.00 | 100 |
| | | 3 | 256 | max | 0.0005 | 16 | 0.10 | 0.00 | 100 |
| | ogbn-arxiv | 3 | 256 | – | 0.0001 | 64 | 0.20 | 0.00 | 200 |
| | | 5 | 256 | – | 0.0001 | 64 | 0.20 | 0.00 | 200 |
| | | 5 | 256 | max | 0.0001 | 64 | 0.20 | 0.00 | 200 |
| | ogbn-products | 3 | 256 | – | 0.001 | 128 | 0.35 | 0.10 | 150 |
| | | 5 | 256 | – | 0.001 | 128 | 0.35 | 0.10 | 150 |
| | | 3 | 256 | max | 0.001 | 128 | 0.35 | 0.10 | 150 |

The hyperparameters to reproduce the Table 1 SHADOW-GNN results are listed in Tables 8 and 9. The hyperparameters to reproduce the Table 2 and 3 SHADOW-GNN results are listed in Table 10.

## E.5 Setup of the SGC Experiments

Following [47], we compute the SGC model as $Y = \texttt{softmax}\left(\widetilde{A}^K X W\right)$ where $\widetilde{A} = \widetilde{D}^{-\frac{1}{2}}\widetilde{A}\widetilde{D}^{-\frac{1}{2}}$ and $\widetilde{A} = I + A$. Matrix $W$ is the only learnable parameter. $K$ is the power on the adjacency matrix and we vary it as $K \in \{1, 3, 5, 10, 20, 30, 40\}$ in the Figure 3 experiments. For SHADOW-SGC, we compute the embedding for target $v$ as $\boldsymbol{y}_v = \left[\texttt{softmax}\left(\widetilde{A}^K_{[v]} X_{[v]} W\right)\right]_{v,:}$.

SGC and SHADOW-SGC are trained with the same hyperparameters (*i.e.*, learning rate equals 0.001 and dropout equals 0.1, across all datasets). SHADOW-SGC uses the same EXTRACT as the SHADOW-GCN model in Table 1. In the legend of Figure 3, due to lack of space, we use S-SGC to denote SHADOW-SGC. We use "F", "R" and "A" to denote the datasets of Flickr, Reddit and ogbn-arxiv respectively.

Table 9: Training configuration of SHADOW-GNN for Table 1 (2-hop EXTRACT)

| Arch. | Dataset | Layers | Dim. | Pooling | Learning Rate | Batch Size | Dropout | DropEdge | Budget $b$ |
|-------|---------|--------|------|---------|---------------|------------|---------|----------|-----------|
| SHADOW-SAGE | Flickr | 3 | 256 | – | 0.0005 | 64 | 0.45 | 0.05 | 20 |
| | | 5 | 256 | – | 0.0005 | 64 | 0.45 | 0.05 | 20 |
| | Reddit | 3 | 256 | – | 0.0001 | 128 | 0.20 | 0.15 | 20 |
| | | 5 | 256 | – | 0.0001 | 128 | 0.20 | 0.15 | 20 |
| | Yelp | 3 | 256 | – | 0.0005 | 16 | 0.10 | 0.00 | 20 |
| | | 5 | 256 | – | 0.0005 | 16 | 0.10 | 0.00 | 20 |
| | ogbn-arxiv | 3 | 256 | – | 0.00005 | 16 | 0.25 | 0.15 | 20 |
| | | 5 | 256 | – | 0.00005 | 16 | 0.25 | 0.15 | 20 |
| | ogbn-products | 3 | 256 | – | 0.002 | 128 | 0.40 | 0.05 | 20 |
| | | 5 | 256 | – | 0.002 | 128 | 0.40 | 0.05 | 20 |

Table 10: Training configuration of SHADOW-GNN for Table 2 and 3 (PPR EXTRACT)

| Dataset | Arch. | Layers | Dim. | Pooling | Learning Rate | Dropout | DropEdge | Budget $p$ | Threshold $\theta$ |
|---------|-------|--------|------|---------|---------------|---------|----------|-----------|-------------------|
| ogbn-papers100M | SHADOW-GAT$_{200}$ | 5 | 800 | max | 0.0002 | 0.30 | 0.10 | 200 | 0.002 |
| | SHADOW-GAT$_{400}$ | 3 | 800 | max | 0.0002 | 0.35 | 0.10 | 400 | 0.002 |
| ogbl-collab | SHADOW-SAGE | 5 | 256 | sort | 0.00002 | 0.25 | 0.10 | 200 | 0.002 |

## F    Additional Experimental Results

### F.1    Understanding the Low Memory Overhead of SHADOW-GNN

SHADOW-GNN in general consumes much less memory than the other GNN-based methods. We can understand the low memory overhead of SHADOW-GNN from two perspectives.

Comparing with the normal GNN model, SHADOW-GNN requires much less GPU memory since both the training and inference of SHADOW-GNN proceeds in minibatches. While other minibatching training algorithms exist for the normal GNNs (*e.g.*, layer sampling [7] and graph sampling [55]), such algorithms either result in accuracy degradation or limited scalability. Therefore, most of the OGB leaderboard methods are tuned with *full-batch* training.

Comparing with the simplified GNN models (*e.g.*, SIGN [11] and SGC [47]), SHADOW-GNN requires much less RAM size. The reason is that both SIGN and SGC rely on preprocessing of node features over the full graph. Specifically, for SIGN, its preprocessing step concatenates the original node features with the smoothened values. For the $(p, s, t)$ architecture of SIGN (see the SIGN paper for definition of $p$, $s$ and $t$), the memory required to store the preprocessed features equals:

$$M = N \cdot f \cdot ((p+1) + (s+1) + (t+1)) \tag{17}$$

where $N$ is the number of nodes in the full graph and $f$ is the original node feature size. For the $(3, 3, 3)$ SIGN / SIGN-XL architecture on the ogbn-papers100M leaderboard (see Table 2), the required RAM size equals $M = 682$GB. This number does not even consider the RAM consumptions due to temporary variables or other buffers for the trainig operations.

For SGC, even though it does not perform concatenation of the smoothened features, it still requires double the original feature size to store the temporary values of $A^K \cdot X$. The original features of ogbn-papers100M takes around 56GB, and the full graph adjacency matrix consumes around 25GB. In sum, SGC requires at least $2 \cdot 56 + 25 = 137$GB of RAM.

Table 11 summarizes the RAM / GPU memory consumption for the leaderboard methods listed in Table 2. Note that our machines do not support the training of SIGN (due to the RAM size constraint), and thus we only show the lower bound of SIGN's RAM consumption in Table 11.

On the other hand, SHADOW-GNN can flexibly adjust its batch size based on the available memory. Even for ogbn-papers100M, a typical low-end server with 4GB of GPU memory and 128GB of RAM is sufficient to train the 5-layer SHADOW-GAT. Increasing the batch size of SHADOW-GNN may further lead to higher GPU utilization for more powerful machines. Figure 7 shows the computation time speedup (compared with batch size 32) and GPU memory consumption for SHADOW-GAT under batch size of 32, 64 and 128. A 5-layer SHADOW-GAT only requires around

Table 11: Memory consumption of the `ogbn-papers100M` leaderboard methods

| Method | CPU RAM | GPU memory |
|---|---|---|
| GraphSAGE+incep | 80GB | 16GB |
| SIGN-XL | >682GB | 4GB |
| SGC | >137GB | 4GB |
| SHADOW-GAT | **80GB** | **4GB** |

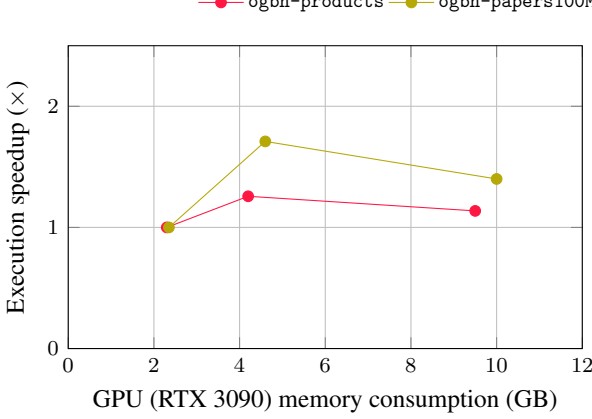

Figure 7: Controlling the GPU memory-speed tradeoff by SHADOW-GAT batch size (32, 64, 128)

5GB of GPU memory to saturate the computation resources of the powerful GPU cards such as NVIDIA RTX 3090.

## F.2 SHADOW-GNN on Other Architectures

In addition to the GCN, GraphSAGE and GAT models in Table 1, we further compare JK-Net and GIN with their SHADOW versions in Table 12. For all the results in Tabel 1, we do not apply pooling or ensemble on SHADOW-GNN. Similar to DropEdge, the skip connection (or "jumping knowledge") of JK-Net helps accuracy improvement on deeper models. Compared with the normal JK-Net, increasing the depth benefits SHADOW-JK more. The GIN architecture theoretically does not oversmooth. However, we observe that the GIN training is very sensitive to hyperparameter settings. We hypothesize that such a challenge is due to the sensitivity of the sum aggregator on noisy neighbors (*e.g.*, for GraphSAGE, a single noisy node can hardly cause a significant perturbation on the aggregation, due to the averaging over the entire neighborhood). The accuracy improvement of SHADOW-GIN compared with the normal GIN may thus be due to noise filtering by the `EXTRACT`. The impact of noises / irrelevant neighbors can be critical, as reflected by the 5-layer GIN accuracy on `Reddit`.

Table 12: Test accuracy on other architectures (PPR `EXTRACT`)

| | Flickr | | Reddit | | ogbn-arxiv | |
|---|---|---|---|---|---|---|
| | Normal | SHADOW | Normal | SHADOW | Normal | SHADOW |
| JK (3) | 0.4945±0.0070 | **0.5317**±0.0027 | 0.9649±0.0010 | **0.9682**±0.0003 | 0.7130±0.0026 | **0.7201**±0.0017 |
| JK (5) | 0.4940±0.0083 | **0.5328**±0.0026 | 0.9640±0.0013 | **0.9685**±0.0006 | 0.7166±0.0053 | **0.7226**±0.0024 |
| GIN (3) | 0.5132±0.0031 | **0.5228**±0.0028 | 0.9345±0.0034 | **0.9578**±0.0006 | 0.7087±0.0016 | **0.7173**±0.0029 |
| GIN (5) | 0.5004±0.0067 | **0.5255**±0.0023 | 0.7550±0.0039 | **0.9552**±0.0007 | 0.6937±0.0062 | **0.7140**±0.0027 |

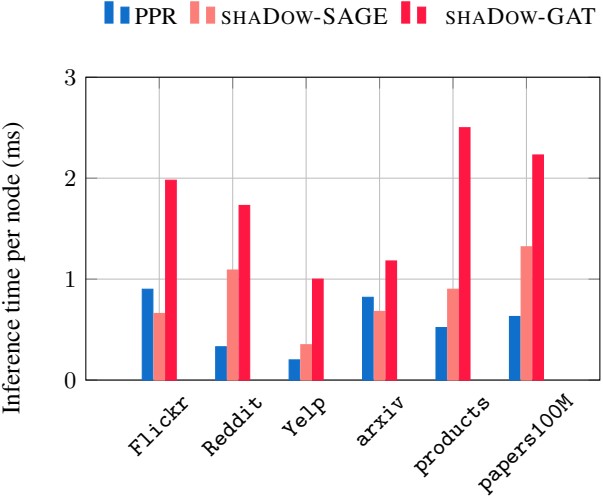

Figure 8: Measured execution time for PPR EXTRACT and the GNN model computation

### F.3 Benefits of Pooling

In Table 13, we summarize the average accuracy of 5-layer SHADOW-SAGE obtained from different pooling functions (see Table 5 for the equations for pooling). For both graphs, adding a pooling layer helps improve accuracy. On the other hand, the best pooling function may depend on the graph characteristics. We leave the in-depth analysis on the effect of pooling as future work.

Table 13: Effect of subgraph pooling on 5-layer SHADOW-SAGE

| Dataset | None | Mean | Max | Sort |
|---|---|---|---|---|
| Flickr | $0.5351\pm0.0026$ | $0.5361\pm0.0015$ | $0.5354\pm0.0021$ | $\mathbf{0.5367}\pm0.0026$ |
| ogbn-arxiv | $0.7302\pm0.0014$ | $0.7304\pm0.0015$ | $\mathbf{0.7342}\pm0.0017$ | $0.7295\pm0.0018$ |

### F.4 Cost of Subgrpah Extraction

We evaluate the PPR EXTRACT in terms of its execution time overhead and accuracy-time tradeoff. In Figure 8, we parallelize EXTRACT using half of the available CPU cores of MACHINE 3 (*i.e.*, 32 cores) and execute the GNN computation on the RTX 3090 GPU. Clearly, the PPR EXTRACT is lightweight: the sampling time is lower than the GNN computation time in most cases. In addition, the sampling time per node does not grow with the full graph size. This shows that SHADOW-GNN is scalable to massive scale graphs. By the discussion in Appendix C, the approximate PPR computation achieves efficiency and scalability by only traversing a local region around each target node.

### F.5 Tradeoff Between Inference Time and Accuracy

To evaluate the accuracy-time tradeoff, we take the 5-layer models of Table 1 as the pretrained models. Then for SHADOW-GNN, we vary the PPR budget $p$ from 50 to 200 with stride 50. In Figure 9, the inference time of SHADOW-GNN has already included the PPR sampling time. Firstly, consistent with Table 1, inference of SHADOW-GNN achieves higher accuracy than the normal GNNs, with orders of magnitude speedup as well. In addition, based on the application requirements (*e.g.*, latency constraint), SHADOW-GNNs have the flexibility of adjusting the sampling size without the need of retraining. For example, on Reddit and ogbn-arxiv, directly reducing the subgraph size from 200 to 50 reduces the inference latency by $2\times$ to $4\times$ at the cost of less than $1\%$ accuracy drop.

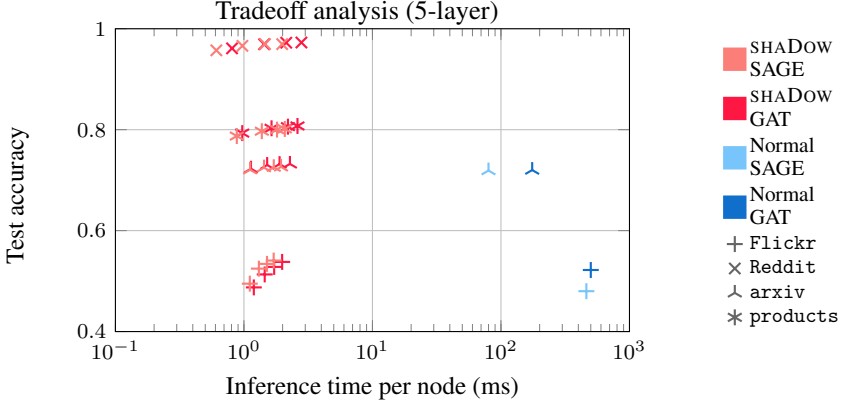

Figure 9: Inference performance tradeoff. We test pre-trained models by subgraphs of various sizes.

Table 14: sHADOW-GCN test accuracy

| $L'$ | EXTRACT | Flickr | ogbn-products |
|------|---------|--------|---------------|
| 3 | PPR | $0.5257 \pm 0.0021$ | $0.7773 \pm 0.0032$ |
| 5 | 2-hop | $0.5210 \pm 0.0023$ | $0.7794 \pm 0.0039$ |
| | PPR | $0.5273 \pm 0.0020$ | $0.7836 \pm 0.0034$ |
| | Ensemble | $\mathbf{0.5304} \pm 0.0017$ | $\mathbf{0.7858} \pm 0.0021$ |
| 7 | PPR | $0.5225 \pm 0.0023$ | $\mathbf{0.7850} \pm 0.0044$ |

## F.6 Ensemble and Deeper Models

Table 14 shows additional results on subgraph ensemble and deeper sHADOW-GCN models. The PPR and 2-hop EXTRACT follow the same configuration as Table 8 and 9. When varying the model depth, we keep all the other hyperparameters unchanged. From both the Flickr and ogbn-products results, we observe that ensemble of the PPR EXTRACT and the 2-hop EXTRACT helps improve the sHADOW-GCN accuracy. From the ogbn-products results, we additionally observe that increasing sHADOW-GCN to deeper than 5 layers may still be beneficial. As analyzed by Figure 2, the model depth of 5 is already much larger than the hops of the subgraph. The 7-layer results in Table 14 indicate future research opportunities to improve the sHADOW-GNN accuracy by going even deeper.