# OpenReview forum: "Decoupling the Depth and Scope of Graph Neural Networks"
_NeurIPS.cc/2021/Conference — NeurIPS 2021 Poster_

### Official Review · Reviewer_ChBa · 2021-07-15

**Rating:** 4
**Confidence:** 4

**Summary:**

The paper introduces a framework to design GNNs with decoupled depth and scope. The main motivation of the paper is to overcome the caveats arising when training GNNs on very large graphs: (i) over-smoothing and (ii) expensive computation; that rise due to the need for a deep GNN to extract meaningful information. The basic idea of the paper is to apply a GNN only on predefined subgraph neighborhoods specified by an $\texttt{EXTRACT}$ operator for each node. The choice of the subgraph neighborhoods defines the GNN scope independently of the depth. The authors explain why this model prevents over-smoothing of the entire graph and shows greater expressivity over commonly used 1-WL GNNs.

**Limitations And Societal Impact:**

The authors do not discuss societal impacts.

**Main Review:**

1. **Originality**

   The authors do not cite the work [1], which suggests a basically very similar idea. [1] could be thought of as a version of this work, where the $\texttt{EXTRACT}$ operator extracts K-hop ego networks and apply GNNs of depth $K$ and not $K’>K$ as suggested in this work. The theoretical result of superior expressivity over 1-WL is also already shown in [1].

   I therefore feel that the novelty of the ideas in this paper is incremental.

2. **Experimental Evaluation**

   It appears that the authors have done an extensive experimental evaluation of their work.

   However, although achieving state-of-the-art performance on most of the datasets, the performance gain is quite small (except for link prediction), and it seems that there are many “tricks” under the hood that were needed to get these results. Furthermore, I would have wanted to see experimental evidence for the need of $L'>L$ layers for a depth-$L$ neighborhood choice. I suspect the empirical benefits do not exist as [1] is able to reach good performance without it.

3. **Theory**

   The authors discuss over-smoothing but then suggest applying a deep GNN on subgraphs which basically over-smooth subgraphs. I wonder whether the authors tried to apply fewer layers?



   Since I’m not convinced by the experimental evaluation, and due to the similarity to [1], I, unfortunately, think that the contribution of this work is limited.

---

[1] You, J., Gomes-Selman, J.M., Ying, R., & Leskovec, J. (2021). Identity-aware Graph Neural Networks. *AAAI*.



**Time Spent Reviewing:**

5

---

> ### Author Response · Authors · 2021-08-10
> **Authors' Response to Reviewer ChBa**
>
> We thank the reviewer for the feedback.
>
> > The authors do not cite the work [1], which suggests a basically very similar idea. [1] could be thought of as a version of this work, where the EXTRACT operator extracts $K$-hop ego networks and apply GNNs of depth $K$ and not $K′>K$ as suggested in this work.
>
> In our opinion, **the two works, ID-GNN [1] and shaDow-GNN, are quite different**. As the reviewer mentioned, ID-GNN applies a $K$-layer GNN on a $K$-hop subgraph. Thus, the receptive field of ID-GNN is exactly the same as that of any other "normal GNNs" (described between lines 110 to 112). So ID-GNN does not decouple the depth and scope of the model. In comparison, one of the key takeaways from our paper is to "decouple" by setting $K’ > K$ (not $K’ = K$ as in ID-GNN).
>
> We will cite ID-GNN in the related work in our next revision.
>
> > The theoretical result of superior expressivity over 1-WL is also already shown in [1].
> > I therefore feel that the novelty of the ideas in this paper is incremental.
>
> Although ID-GNN and shaDow-GNN achieve the same outcome as surpassing 1-WL, **the approaches are completely different**. The key idea of ID-GNN is to augment node features with node identity, while the key idea of shaDow-GNN is to decouple the depth and scope of the GNN. Thus, even though we have already known message-passing GNNs can surpass 1-WL, the shaDow-GNN analysis still brings much value to the community by revealing a *new mechanism* to achieve the goal. Note that our decoupling mechanism is scalable to very large graphs.
>
> Secondly, we would like to emphasize that our contributions are much broader than the 1-WL related analysis. Throughout the paper, we have
> 1. proposed a first-of-its-kind “decoupling” design principle for the general GNN architecture;
> 2. theoretically justified expressive power improvement from three different perspectives (Sections 3.1, 3.2, 3.3);
> 3. proposed algorithmic guidelines for implementing such a principle and two example algorithms (Section 3.4);
> 4. proposed novel architecture extensions uniquely enabled by the principle (Section 3.5).
>
> Thus, it may not be fair to judge the overall novelty only from the 1-WL related analysis, and ignore the other contributions.
>
> > although achieving state-of-the-art performance on most of the datasets, the performance gain is quite small (except for link prediction)
>
> **Performance gains are significant**, w.r.t. all three metrics: accuracy, efficiency and hardware requirements.
>
> * *Accuracy*: From Table 1 (node classification), the shaDow versions achieve significant accuracy gain compared to the corresponding normal versions. There do exist exceptions (e.g., on ogbn-products, GCN+GraphSAINT vs. shaDow-GCN+PPR). But the overall trend clearly shows the improvement is significant. Some examples:
>     * For Flickr, 0.5201 (GraphSAGE+GraphSAINT) → 0.5424 (shaDow-SAGE)
>     * For ogbn-arxiv, 0.7201 (GAT) → 0.7283 (shaDow-GAT)
>     * For ogbn-products, 0.7964 (GraphSAGE+GraphSAINT) → 0.8043 (shaDow-SAGE)
>     * …
> * *Efficiency*: The computational advantages are *huge* -- as can be seen from Table 1 and 2. Especially, for the largest ogbn-papers100M in Table 2, shaDow-GNN reduces the neighborhood size by three orders of magnitude.
> * *Hardware requirements*: shaDow-GNN dramatically reduces the memory consumption for both CPU and GPU. Due to the space constraints in the main paper, we have presented detailed memory study in Table 11 and Figure 6 in Appendix. Even for the gigantic ogbn-papers100M graph, both the training and inference of shaDow-GNN can be run on a low-end server (with 80GB RAM) and a low-end GPU (with 4GB memory).
>
> > it seems that there are many “tricks” under the hood that were needed to get these results
>
> We would like to emphasize that **there are no “tricks” to generate the results in the experiment section**. To ensure fair comparison and facilitate reproduction of results, we have
> * Clearly summarized the critical experimental setups in the “Setup” paragraph of Section 5 in the main text.
> * Described in fine detail all the other experimental configurations in Appendix, including
>     * Appendix E.1: Dataset details
>     * Appendix E.2: Hardware specifications
>     * Appendix E.3: Software specifications
>     * Appendix E.4: Hyperparameter tuning procedures, and the hyperparameter values (see Tables 8, 9, 10) corresponding to the Table 1, 2, 3 results
> * Provided the full codebase in supplementary materials, which includes
>     * Step-by-step guide for compilation and execution
>     * Software version and environments
>     * Recommended hardware capacity
>     * Troubleshooting for common compilation issues
>
> We are more than happy to further clarify the setup if the reviewer still finds something confusing.
>
> > I would have wanted to see experimental evidence for the need of $L′>L$ layers for a depth-$L$ neighborhood choice.
>
> In Table 1, we have already shown the empirical evidence that setting $L’ > L$ can lead to accuracy gains for a depth-$L$ neighborhood. Note that according to Figure 1, the PPR subgraph can be seen as of depth-3 (i.e., $L=3$) for Flickr, Yelp and ogbn-arxiv. Comparing the rows of shaDow $L’=3$ and the rows of shaDow $L’=5$, we observe that in many cases, increasing the layers from 3 to 5 leads to accuracy gains. For example:
> * For Flickr, 0.5344 (shaDow-SAGE, $L’=3$) → 0.5424 (shaDow-SAGE, $L’=5$)
> * For Yelp, 0.5255 (shaDow-GCN, $L’=3$) → 0.5272 (shaDow-GCN, $L’=5$)
> * For ogbn-arxiv, 0.7243 (shaDow-GAT, $L’=3$) → 0.7283 (shaDow-GAT, $L’=5$)
> * …
>
> In addition, during rebuttal, we further evaluated the accuracy of the 2-layer baseline models. Using the GCN / GraphSAGE / GAT architectures on our 5 graphs, we observe that a 2-layer baseline cannot outperform a 3-layer baseline w.r.t. accuracy. This implies that, the current Table 1 results already reflect the best baseline accuracy under $L=L'=2$, $3$ or $5$. Therefore, Table 1 provide convincing evidence that “decoupling the depth and scope” leads to significant accuracy gains.
>
> > I suspect the empirical benefits do not exist as [1] is able to reach good performance without it. ”
>
> **The performance of ID-GNN and shaDow-GNN are unrelated.**
> * There is not any common dataset between ID-GNN and shaDow-GNN.
> * ID-GNN and shaDow-GNN use completely different techniques for improving expressive power.
>
> > The authors discuss over-smoothing but then suggest applying a deep GNN on subgraphs which basically over-smooth subgraphs. I wonder whether the authors tried to apply fewer layers?
>
> Addressing oversmoothing in **deep GNNs** is exactly a key motivation for shaDow-GNN. Section 3.1 shows that local smoothing in the subgraph will not wipe out the feature and structure information around the target node. Thus, we can flexibly adjust the depth of shaDow-GCN without worrying about oversmoothing (in the global graph). In summary, we didn’t try to apply fewer layers.
>
> Please let us know if there still remains unresolved concerns. Thanks!

---

### Official Review · Reviewer_29rS · 2021-07-16

**Rating:** 7
**Confidence:** 3

**Summary:**

This paper aims to tackle two challenges in GNN: oversmoothing and neighbor explosion. To this end, the authors introduce a model, SHADOW, to decouple GNN's depth (number of layers) and scope (receptive field). The proposed model enables GNNs to treat their depth and scope independently, leading to provable improved capacity for GCN, GraphSAGE, and GIN. Extensive experiments show SHADOW's clear advantage over state-of-the-art methods.

**Limitations And Societal Impact:**

Yes, the authors have adequately addressed the limitations and potential negative societal impact of their work.

**Main Review:**

Strength:
* Connections to existing works are well established. Section 3 describes the connection between SHADOW and GCN/GraphSAGE/GIN, and how SHADOW can improve these GNNs. Theoretically, SHADOW can: (1) help GCN to alleviate oversmoothing, (2) extend GraphSAGE to approximate functions while GraphSAGE solely can not, and (3) enable GIN to go beyond 1-WL.
* After decoupling, the depth and scope of GNN can be tuned independently. Notably, the authors explain why we need a larger depth than a larger scope. As an example, SHADOW-SAGE benefits from using a deep model, which may have larger depth than scope.
* The design of subgraph extraction functions is well studied.
* The design of experiments is reasonable, involving various datasets. The choices of baselines are fair.
Sufficient ablation studies validate the model choices (e.g., six GNNs are examined as model's backbone).
* The paper is well written and easy to follow. Code is provided for reproducible purposes.

I did not see outstanding weaknesses in this paper. Two recommendations:
* It is better to prioritize SHADOW-GIN results to the main text instead of appendix. Because one significant contribution is that SHADOW improves GIN to exceed the power of 1-WL.
Besides, empirical evaluation on SHADOW-GIN's ability to detect graph isomorphism can be considered.

* For some graphs, distant neighbor matters, which could be a limitation for the proposed model. Though this paper claims many real-world graphs exhibit minor impacts on distant nodes, but mentioning such a limitation in the main paper is recommended.

**Time Spent Reviewing:**

4

---

> ### Author Response · Authors · 2021-08-10
> **Authors' Response to Reviewer 29rS**
>
> We appreciate the positive feedback of the reviewer.
>
> > It is better to prioritize SHADOW-GIN results to the main text instead of appendix. Because one significant contribution is that SHADOW improves GIN to exceed the power of 1-WL. Besides, empirical evaluation on SHADOW-GIN's ability to detect graph isomorphism can be considered.
>
> Thanks for the suggestion. We agree that it may better highlight the results if the shaDow-GIN experiments are placed in the main text. We will re-organize the experiment section in the next revision. We will also consider adding synthetic datasets for evaluating the ability of graph isomorphism test.
>
> > For some graphs, distant neighbor matters, which could be a limitation for the proposed model. Though this paper claims many real-world graphs exhibit minor impacts on distant nodes, but mentioning such a limitation in the main paper is recommended.
>
> We agree that in some cases, distant neighbors may play an important role. In the next revision, we will clarify that our paper focuses on graphs such as citation networks, social networks and product recommendation networks, where a shallow neighborhood contains the most critical information. We will add a "limitation" section in the main paper.
>
> Thanks again for your feedback!

---

### Official Review · Reviewer_KB4T · 2021-07-18

**Rating:** 5
**Confidence:** 3

**Summary:**

This work proposes a design principle (SHADOW) to enrich the expressive power of GNNs: it first extracts a localized subgraph as the fixed input receptive field to the GNN, and then the depth of the GNN can be selected separately. This work shows that the proposed design can be applied to various GNNs (e.g., SHADOW-GCN, SHADOW-SAGE, and SHADOW-GAT), and it provides theoretical analysis on the expressive power of three SHADOW-GNNs (the ones mentioned above) comparing to their original counterparts. It further conducts experiments on several large graph benchmark datasets and demonstrates the superior performance of SHADOW-GNNs.

**Limitations And Societal Impact:**

The authors do not discuss any potential negative societal impacts of their work. Some possible perspectives to think about potential societal impacts include energy consumption, reusability of trained models, easy deployment on mobile devices and so on.

**Main Review:**

This work proposes a "decoupling" design principle that separates the depth and scope of GNNs. The work provides some theoretical and empirical support to demonstrate the proposed design improves both the accuracy and computation efficiency. To be more specific, the work provides theoretical analysis on the expressive power of the proposed design, and hand-constructed cases where the SHADOW-GNNs are more expressive than its standard counterparts. The experimental results also show some improvements of the SHADOW-GNNs in test accuracies on ogbn benchmark datasets (6 node classification datasets and 1 link prediction dataset). The presentation of the work is very clear, and I enjoy reading it through. This work also tries to tackle two fundamental obstacles in deep GNNs: the expressive challenge and the computation, which are significant problems to be addressed in the community.

While the work studies an important problem, my main concern lies in the EXTRACT function of the proposed principle.
1. It seems most theorems and corollaries presented in this work are highly dependent on some appropriate EXTRACT functions (e.g., Corollary 3.2.1, Corollary 3.2.2., and Theorem 3.3). In these cases, the EXTRACT function needs to generate a localized subgraph for each of the nodes to satisfy certain properties/constraints, but it is not clear how we can obtain such an EXTRACT function without too much prior information on the given tasks. As the success of a SHADOW-GNN is greatly dependent on a good EXTRACT function, as a design principle, it is also vital to provide clear guidance on how to find/design the EXTRACT function.
2. Although the authors slightly touch on 3 possible directions to come out with appropriate EXTRACT functions: heuristic-based, model-based, and learning-based, all of them still lack practical guidance. The model-based approaches need access to the generation process of the input graphs, which might not be available in real-world tasks. The authors also provide an example of a heuristic-based approach, which is based on a hand-engineered feature (the page rank score of other nodes to the given node), and such graph metrics are often based on human insights and expertise. The learning-based approach sounds interesting but is left to future work.
3. How does the proposed design principle differ from neighbor-sampling (e.g. [1, 2]) and subgraph-sampling approaches (e.g. [3, 4])?

Minor comment:
- provide a more direct comparison for the reduction in computation and hardware cost with their counterparts: e.g., add new legends to Figures 6 and 7 representing the performance of the standard GNNs.

[1] William L Hamilton, Zhitao Ying, and Jure Leskovec. Inductive representation learning on large graphs. In NeurIPS, 2017.

[2] Jianfei Chen, Jun Zhu, and Le Song. Stochastic training of graph convolutional networks with variance reduction. In ICML, 2018.

[3] Wei-Lin Chiang, Xuanqing Liu, Si Si, Yang Li, Samy Bengio, and Cho-Jui Hsieh. Cluster-gcn: An efficient algorithm for training deep and large graph convolutional networks.  In Proceedings of the 25th ACMSIGKDD International Conference on Knowledge Discovery & Data Mining, pages 257–266, 2019.

[4] Hanqing Zeng, Hongkuan Zhou, Ajitesh Srivastava, Rajgopal Kannan, and Viktor Prasanna. Graphsaint: Graph sampling based inductive learning method. In ICLR, 2019.

**Time Spent Reviewing:**

8

---

> ### Author Response · Authors · 2021-08-10
> **Authors' Response to Reviewer KB4T**
>
> We appreciate the constructive feedback from the reviewer.
>
> > most theorems and corollaries presented in this work are highly dependent on some appropriate EXTRACT functions ...
> > the EXTRACT function needs to generate a localized subgraph for each of the nodes to satisfy certain properties/constraints
>
> The constraints on EXTRACT are in fact mild. Specifically, the constraint by Corollary 3.2.1 is that subgraph sizes are bounded. The constraint by Corollary 3.2.2 is that the node set of $u$’s subgraph is not identical to the node set of $v$’s subgraph. The constraint by Section 3.2 is that subgraph node features are relevant to the target node. The constraint by Theorem 3.3 is that "a subgraph of a regular graph may not be regular" (see line 219).
>
> For example, consider the PPR EXTRACT used in our experiments. The extraction function returns neighbors with top-$p$ PPR scores (lines 770, Appendix C). And so such PPR EXTRACT belongs to EXTRACT_1. In addition, PPR EXTRACT can be empirically regarded as belonging to EXTRACT_2 since the function returns distinct neighborhood in most cases (for conciseness of response, we omit the detailed profiling data here. We are more than glad to expand this point if the reviewer asks for it).
>
> > it is not clear how we can obtain such an EXTRACT function without too much prior information on the given tasks
>
> Using domain knowledge is a common and reasonable way to improve learning performance. For example, in APPNP [22], GDC [23] and PPRGo [4], the PPR algorithm is also explicitly utilized (although in a different way from us). More importantly, for many types of graphs (e.g., social network as evaluated in our experiments), we already have rich "prior information" thanks to the long-lasting graph analytics literature.
>
> > it is also vital to provide clear guidance on how to find/design the EXTRACT function
>
> We have presented three guidelines in Section 3.4 and Appendix C. See more clarification below.
>
> > the authors slightly touch on 3 possible directions to come out with appropriate EXTRACT functions
>
> The core of the paper is "decoupling" as a general design principle, rather than the many different ways to implement such a principle. The 3 possible directions can lead to numerous algorithm variants, and we think it unnecessary to expand them in the main text. For readers' interest, we have discussed details of the 3 directions in Appendix C.
>
> > The model-based approaches need access to the generation process of the input graphs, which might not be available in real-world tasks
>
> We agree that the ground-truth generation process may not be available in practice. However, it has been widely shown that graphs from certain hypothetical generation process can approximate realistic data very well [a, b].
>
> > The authors also provide an example of a heuristic-based approach, which is based on a hand-engineered feature
>
> PPR is a classic algorithm widely used to discover interesting neighbors. There is rich literature analyzing the graph theoretical properties of PPR. In our opinion, PPR is not a hand-engineered feature.
>
> > such graph metrics are often based on human insights and expertise
>
> We agree. And our proposed decoupling principle provides an easy way to efficiently utilize such human insights to enhance any GNN architecture.
>
> > The learning-based approach sounds interesting but is left to future work
>
> We have provided an example of the learning based approach in Appendix C. The example describes the detailed training algorithm (lines 789 to 796). However, we would like to emphasize that our focus is the design principle itself, rather than the various possible implementations of the principle.
>
> > How does the proposed design principle differ from neighbor-sampling (e.g. [1, 2]) and subgraph-sampling approaches (e.g. [3, 4])?
>
> Our shaDow-GNN is fundamentally different from sampling approaches [1, 2, 3, 4]. We have described their differences in Sections 1 and 4. We summarize the key takeaways here:
>
> * Despite both changing the receptive field, sampling approaches aim to efficiently approximate some quantities related to the full graph (such as approximating the aggregation of full $L$-hop neighborhood), while shaDow-GNN inherently does not aim at estimating any full-graph quantities because it is designed to look at only locally related subgraph. This conceptually distinguishes shaDow-GNN from sampling approaches (*including both neighbor and graph sampling*).
> * Sampling approaches can only accelerate the training computation, while shaDow-GNN accelerates both the training and inference. Training (multiple epochs) is more tolerant to the information loss from sampling, while inference generates the predictions only once. That’s why practical implementations of sampling approaches apply sampling only during training, but not during inference (in order to avoid accuracy loss). In contrast, shaDow-GNN applies the same subgraph extraction algorithm on both training and test data. This explains why shaDow-GNN achieves significant reduction in inference computation cost, even compared with sampling based models such as GraphSAINT (see Table 1).
> * Sampling approaches do not decouple the depth and scope of GNNs -- the receptive field of an $L$-layer sampling based GNN is still (on expectation) the full $L$-hop neighborhood. In contrast, the key insight of shaDow-GNN is to decouple the depth and scope, which allows using $L’$ layer GNNs on the $L$-hop neighborhood ($L’>L$).
>
> > provide a more direct comparison for the reduction in computation and hardware cost with their counterparts: e.g., add new legends to Figures 6 and 7 representing the performance of the standard GNNs.
>
> We will take the advise and do so in our next revision.
>
> References
>
> [a]: Leskovec et al., Graph Evolution: Densification and Shrinking Diameters
>
> [b]: Leskovec et al., Sampling from Large Graphs
>
> Again, we thank the reviewer for the comments. Please let us know of any unresolved concerns.

---

> > ### Comment · Reviewer_KB4T · 2021-09-04
> > **Response to Authors**
> >
> > Thanks for the authors' response. While the response clarified some of my concerns, I am still a bit concerned about the EXTRACT function as well as the assumptions in theoretical results. For example, it is still unclear how to construct an EXTRACT function satisfying a given property/constraint, and how do we know if such EXTRACT function exists with reasonable computational complexity. I believe more investigations on analyzing the limitations of various types of EXTRACT functions as well as obtaining/learning good EXTRACT functions for a given task would make this work into a better shape for future re-submission.

---

> > > ### Author Response · Authors · 2021-09-04
> > > **Thanks and Response**
> > >
> > > Thank you for the response! We are glad that some of the reviewer’s concerns have been addressed. However, since the reviewer’s response has only been made visible to us on Sept 3 (when the rolling discussion phase has ended), we would like to ask the AC / reviewer to please consider our follow-up when making a decision.
> > >
> > > Regarding the concerns that still remain:
> > >
> > > ### Focus of our paper
> > > Our paper introduces a new dimension (i.e., the EXTRACT function) into the GNN design space. We first propose / define this new dimension, and then justify why including this dimension is beneficial. The focus of this paper is our expanded design space, rather than the specific design points (i.e., specific EXTRACT functions). Therefore, even though we agree that investigating more EXTRACT functions is interesting, it would deviate from the main theme of our paper.
> > >
> > > ### Feasibility of example EXTRACT
> > > On the other hand, following our guidelines, we have proposed and evaluated two example EXTRACT functions: PPR and k-hop. **Both EXTRACTs satisfy the constraint of our theorems**. **Both EXTRACTs have low computation complexity**.
> > >
> > > * The k-hop EXTRACT can be used to prove Theorem 3.3 (see detailed steps in Appendix A).
> > >
> > > * Both the k-hop and PPR EXTRACTs can support the shaDow-SAGE analysis in Section 3.2.
> > >
> > > * The PPR EXTRACT belongs to the EXTRACT functions defined by Corollary 3.2.1 (see our initial response).
> > >
> > > * The PPR EXTRACT empirically belongs to the EXTRACT function defined by Corollary 3.2.2. As we mentioned, we are more than happy to provide the empirical evidence. Please see the following:
> > >
> > > *Empirical evidence for PPR EXTRACT*: For Corollary 3.2.2, EXTRACT needs to return non-identical subgraphs for different target nodes. We additionally perform the following experiment. Given a graph, we randomly pick a target node $v$ and obtain its $G_{[v]}$. We then compare $G_{[v]}$ with $G_{[u]}$ for all $u\neq v$ in the graph. We report on expectation, how many $G_{[u]}$ are identical to $G_{[v]}$. For the PPR based EXTRACT following Table 8’s hyperparameters, the expected number of $u$ such that $G_{[u]}=G_{[v]}$ is: 0.0155 (Reddit), 0.219 (ogbn-arxiv) and 0.1235 (ogbn-products). This means, on ogbn-products of millions of nodes, only less than 1 (0.1235 in expectation) pair of nodes will have the same subgraph, which is negligible.
> > >
> > > ### Summary
> > >
> > > **We believe the above clearly shows how to construct a computationally efficient EXTRACT function satisfying the constraints of our theorems**.
> > >
> > > In our future work, we would love to analyze the limitations of more and more EXTRACT functions, and develop algorithms for automatically learning good EXTRACT.
> > >
> > > Please let us know if your concerns still remain. Thanks again for the valuable discussion!

---

### Official Review · Reviewer_3DBS · 2021-07-25

**Rating:** 6
**Confidence:** 4

**Summary:**

This paper proposes to control the receptive field of GNNs by extracting a small subgraph for every node, e.g. a 2-hop neighborhood. This has computational advantages since it is now easy to mini-batch nodes with their corresponding subgraphs and it allows to build deeper GNNs acting independently on each subgraph, thus avoiding oversmoothing. The paper demonstrated empirical improvements on several node classification benchmarks and a link prediction task.

**Limitations And Societal Impact:**

The authors do not discuss the limitations of their approach. The main limitation in my opinion is the strong assumption that "a shallow neighborhood is necessary and sufficient for the GNN to learn well" - this is not true in the problems where long-range dependencies are important, and where GNNs are currently underperforming due to oversmoothing. Please see the main review for more details.

**Main Review:**

- The proposed approach has computational advantages and leads to small empirical improvements, however, the main motivation is unclear. As the authors state, using $L$-hop subgraphs is equivalent to a GNN with $L$ layers. ShadowGNN allows to build models with $L' > L$ layers while keeping the receptive field to $L$ hops. I am not convinced that this is beneficial: while using more layers can be beneficial as it increases the number of parameters and overall depth, one can alternatively use a $L$-layer GNN with deeper MLPs (or ResNets, etc.) at every layer. I think it is important to conduct additional experiments to show that setting $L' > L$ in ShadowGNN is indeed beneficial as opposed to the aforementioned way of deepening GNNs. Currently, it is unclear, neither theoretically nor empirically, what is the benefit of allowing nodes to communicate with each other multiple times. In the datasets considered in experiments, since 2-hop neighborhood is sufficient to capture important information in the graph, I'd be interested to see a comparison to 2-layer GNNs with deeper feature extractors, i.e. $H^{(l)}$ can be a deeper MLP/ResNet to match the number of parameters of a $L'=5$ ShadowGNN.
- "a shallow neighborhood is necessary and sufficient for the GNN to learn well" - this is a very strong claim and I do not think that it is true. I recommend revising it and adding a discussion of the limitations of the proposed method in the context of capturing long-range dependencies. While a shallow neighborhood may be sufficient for social and citation network graphs, it could be insufficient to capture important properties of molecules in chemistry applications. The problem of designing GNNs that can capture long-range dependencies and not suffer from oversmoothing was studied in "On the bottleneck of graph neural networks and its practical implications" (2021) by Alon and Yahav.
- l342 says that $L' > 5$ can further improve accuracy, but the performance of $L' = 7$ in Table 14 in the appendix does not show any improvements.
- Could you please clarify what you mean by "a depth-3 $G_{[v]}$ differs from the 3-hop neighborhood$ in l359.

Minor:
- l63: "the a"
- l91: missing square brackets in the index
- l114: "sufficeint" -> sufficient
- l245: "we would a" - missing a verb

**Time Spent Reviewing:**

5

---

> ### Author Response · Authors · 2021-08-10
> **Authors' Response to Reviewer 3DBS**
>
> We thank the reviewer for the constructive and valuable feedback.
>
> > The proposed approach has computational advantages and leads to small empirical improvements
>
> Empirical improvements in accuracy, computation and memory consumption are **all significant**:
> * *Accuracy*: From Table 1, the shaDow version achieves significant accuracy gain compared to the corresponding normal version. There do exist exceptions (e.g., on ogbn-products, GCN+GraphSAINT vs. shaDow-GCN+PPR). However, the overall trend shows the significance of the improvement. Some examples:
>     * For Flickr, 0.5201 (GraphSAGE+GraphSAINT) → 0.5424 (shaDow-SAGE)
>     * For ogbn-arxiv, 0.7201 (GAT) → 0.7283 (shaDow-GAT)
>     * For ogbn-products, 0.7964 (GraphSAGE+GraphSAINT) → 0.8043 (shaDow-SAGE)
>     * …
> * *Computation*: The computational advantages are huge -- as can be seen from Table 1 and 2. Especially, for the largest ogbn-papers100M in Table 2, shaDow-GNN reduces the neighborhood size by three orders of magnitude.
> * *Memory*: shaDow-GNN dramatically reduces the memory consumption for both CPU and GPU. Due to the space constraints in the main paper, we have presented detailed memory study in Table 11 and Figure 6 in Appendix. Even for the gigantic ogbn-papers100M graph, both the training and inference of shaDow-GNN can be run on a low-end server (with 80GB RAM) and a low-end GPU (with 4GB memory).
>
> > while using more layers can be beneficial as it increases the number of parameters and overall depth, one can alternatively use a L-layer GNN with deeper MLPs (or ResNets, etc.) at every layer
>
> More parameters and depth alone are not enough, the key operation benefiting expressive power is the *additional message passing*. See more discussion below.
>
> > Currently, it is unclear, neither theoretically nor empirically, what is the benefit of allowing nodes to communicate with each other multiple times.
>
> In Sections 3.1, 3.2 and 3.3, we have shown various theoretical evidence that performing more message passings in the neighborhood is the key to improve the expressive power of shaDow-GNN. Since MLP layers do not perform any node-to-node message passing, adding MLP layers cannot achieve the benefits discussed in Sections 3.1, 3.2 and 3.3.
>
> Theorem 3.3 provides a clear illustration. In the theorem, $f_1$ and $f_2$ can be an arbitrary function, and we have stated that they can be implemented by MLPs (with any depth). Therefore, **an $L$-layer model following Equation 2 can exactly express the design (with deep MLP) proposed by the reviewer**. Now, from the conclusion of Theorem 3.3, the higher expressive power exactly comes from the additional message passings of the $L’$ layers. Figure 4 (Appendix A.2) provides an example graph to illustrate Theorem 3.3, where the red and blue node can only be discriminated by an additional message passing layer (not an additional MLP layer).
>
> Sections 3.1 and 3.2 provide two additional perspectives to justify the theoretical benefits of more message passings. In Section 3.1, different numbers of shaDow-GCN message passings correspond to different levels of smoothing on the local graph signal. The optimal level of smoothing can be task-specific. In Section 3.2, the additional message passings make shaDow-SAGE a better function approximator, where the approximation error only vanishes when the number of message passing iterations goes to infinity.
>
> For the empirical evidence, please see below.
>
> > I think it is important to conduct additional experiments to show that setting $L′>L$ in ShadowGNN is indeed beneficial as opposed to the aforementioned way of deepening GNNs.
>
> We have performed additional experiments to evaluate the effect of deeper MLPs. For the $L’=3$ shaDow-GNN model (PPR EXTRACT), we add two more MLP layers after the last GNN layer (the overall architecture is denoted by "3+2"). The hidden dimensions of the MLP layers are set to be the same as those of the GNN layers. We evaluate all the shaDow-GNN variants in Table 1 (i.e., shaDow-GCN, shaDow-SAGE, shaDow-GAT), and find that the "3+2" architectures produce almost identical accuracy as the "3" architectures -- the largest accuracy difference is less than 0.0007. Since it is empirically clear that adding deeper MLPs has negligible impact on accuracy, here we omit the detailed experiment table for conciseness of response.
>
> > I'd be interested to see a comparison to 2-layer GNNs with deeper feature extractors
>
> We have performed additional experiments to evaluate shaDow-SAGE (PPR EXTRACT) under $L’=2$. Similar to the above configuration, we add three additional MLP layers to construct a "2+3" architecture. For Flickr, ogbn-arxiv and ogbn-products, it is clear that increasing the number of GNN layers (i.e., increasing the number of message passings) helps improve accuracy.
>
> |     | Flickr | Reddit | Yelp | ogbn-arxiv | ogbn-products |
> | --- | ------- | -------- | ------ | -------------- | ------------------- |
> | shaDow-SAGE (2+3) | 0.5273 | 0.9680 | 0.6519 | 0.7204 | 0.7946 |
> | shaDow-SAGE (5) | 0.5424 | 0.9691 | 0.6502 | 0.7255 | 0.8043 |
>
> > "a shallow neighborhood is necessary and sufficient for the GNN to learn well" - this is a very strong claim and I do not think that it is true. I recommend revising it and adding a discussion of the limitations of the proposed method in the context of capturing long-range dependencies.
>
> We agree that the scope of the paper can be better specified. We will state at the beginning of Section 3 that we consider graphs where shallow neighborhoods contain the most critical information. Such graphs include citation networks, social networks and product recommendation networks. We will also add a "limitation" section, saying that shallow neighborhoods may not be sufficient for graphs with long-range dependencies, and cite the mentioned work by Alon and Yahav (2021).
>
> > The problem of designing GNNs that can capture long-range dependencies and not suffer from oversmoothing was studied in "On the bottleneck of graph neural networks and its practical implications" (2021) by Alon and Yahav.”
>
> We would like to point out that Alon and Yahav (2021) studies "over-squashing" rather than "over-smoothing". The two are different phenomena. In their own words of Alon and Yahav: *"We hypothesize that in long-range problems, the explanation for the degraded performance is over-squashing rather than over-smoothing".*
>
> > l342 says that $L′>5$ can further improve accuracy, but the performance of $L′=7$ in Table 14 in the appendix does not show any improvements.
>
> In Table 14, for ogbn-product, the "$L’=7$, PPR" setting achieves slightly higher accuracy than the "$L’=5$, PPR" setting (i.e., 0.7850 vs 0.7836). Note that the variance of the $L’=7$ accuracy is also larger -- there can be other challenges in optimizing a deep GNN such as vanishing gradient. Such challenges exist in general deep neural networks (not limited to the GNN-type). And addressing them is beyond the scope of this work.
>
> Note that we put the $L’=7$ results in the Appendix, because results on $L’=3$ and $L’=5$ have already clearly shown the benefits of our decoupling principle.
>
> > Could you please clarify what you mean by "a depth-3 G[v] differs from the 3-hop neighborhood in l359.
>
> Note that shaDow-GNN is broader than "using an $L’$-layer GNN on an $L$-hop neighborhood". "$L$-hop" is only an example EXTRACT function, and shaDow-GNN admits many other EXTRACT designs (see discussion in Section 3.4).
>
> To describe a general neighborhood, we introduce the concept of "depth" (not "hop") of the subgraph. See Definition 2.1 of Section 2.
> For example, if a node has $10$ 1-hop neighbors, $100$ 2-hop neighbors, and $1000$ 3-hop neighbors, then the "3-hop neighborhood" will include the $10 + 100 + 1000$ nodes. On the other hand, the EXTRACT function can define "a depth-3 $G_{[v]}$" very differently -- e.g., $G_{[v]}$ can just have $10 + 10 + 10$ nodes, where $10$ nodes are from the 1-hop neighbors, other $10$ nodes are selected from all the $100$ 2-hop neighbors, and the rest $10$ nodes are selected from all the $1000$ 3-hop neighbors.
>
> The difference between the depth-3 $G_{[v]}$ and the 3-hop neighborhood is observed in Figure 1.
>
> We thank the reviewer again for the feedback. Please let us know of any unresolved concerns.

---

### Decision · Program_Chairs · 2021-09-28

**Decision:**

Accept (Poster)

**Comment:**

This paper presents an interesting idea of limiting GNNs to local neighborhoods around each node, which can be used to decouple the depth and scope dimensions of the model.  This change led to great empirical results and computational savings.

However, on the theoretical side the authors made some very strong claims without proper support.  To name a few:
* “A shallow neighborhood is necessary and sufficient for the GNN to learn well.”  This is clearly over-claiming.  Lots of problems that require global reasoning across a whole graph cannot be solved when restricting to small neighborhoods.
* “proposed a first-of-its-kind “decoupling” design principle for the general GNN architecture;” (in discussions)  This is over-claiming.  It is not unusual to see GNNs using more layers than the graph diameter.  For example [this paper](https://arxiv.org/abs/1802.03685) even used GNNs up to 1000-layers to solve certain problems.

Also after a closer inspection of the proofs of the various claims in the paper, the AC found that some proofs are questionable, or at least unclear.

For example, in the proof for shaDow-SAGE expressivity: in order to simulate a L-layer GraphSAGE, the EXTRACT function needs to return an L-hop neighborhood.  The proof notably does not cover cases where the neighborhood size is smaller than L which is the case targeted by the paper, and it is obvious that some functions that can be represented by L-layer GraphSAGE won’t be able to be represented by the proposed Shadow-SAGE with a neighborhood size < L.

Similarly, it is clear that there are some graphs distinguishable by shaDow-SAGE with a small neighborhood not distinguishable by 1-WL.  But on the other hand the provided proof does not cover cases where some graphs may be distinguished by 1-WL but potentially can not be distinguished by shaDow-SAGE with a small neighborhood size.

For these proofs it would be good to clarify when we are talking about shaDow-SAGE with a small neighborhood size (the most important and interesting case for this paper), and when we are talking about shaDow-SAGE with a full graph as the neighborhood size (not interesting and not even relevant for the main discussion of this paper).

I hope the authors can improve the writing of this paper based on the feedback and suggestions, and send the paper to the next venue.

AC

**Consistency Experiment:**

NeurIPS has a long history of experimentation. In 2014, NeurIPS ran an experiment in which 10% of submissions were reviewed by two independent committees to quantify the randomness in the review process. This year, we repeated a variant of this experiment to see how the quality of the review process has changed over time.  This paper was part of the experiment and was therefore assigned to two committees (consisting of reviewers, an Area Chair, and a Senior Area Chair) that reached independent decisions.  If both committees made the same recommendation, this recommendation was followed. If a single committee recommended acceptance, the paper was accepted (with the exception of a few cases in which the other committee identified what we considered a fatal flaw, e.g., an error in a key result).

This copy’s committee reached the following decision: **Reject**

The other committee assigned to the paper recommended **Accept (Poster)**.  You can find the other set of reviews, along with any follow up discussion with the authors here:
https://openreview.net/forum?id=_IY3_4psXuf